# Conditional deletion of *Nedd4-2* in lung epithelial cells causes progressive pulmonary fibrosis in adult mice

Julia Duerr [ID] et al.#

Idiopathic pulmonary fibrosis (IPF) is a chronic progressive interstitial lung disease characterized by patchy scarring of the distal lung with limited therapeutic options and poor prognosis. Here, we show that conditional deletion of the ubiquitin ligase *Nedd4-2* (*Nedd4l*) in lung epithelial cells in adult mice produces chronic lung disease sharing key features with IPF including progressive fibrosis and bronchiolization with increased expression of Muc5b in peripheral airways, honeycombing and characteristic alterations in the lung proteome. NEDD4-2 is implicated in the regulation of the epithelial Na+ channel critical for proper airway surface hydration and mucus clearance and the regulation of TGFβ signaling, which promotes fibrotic remodeling. Our data support a role of mucociliary dysfunction and aberrant epithelial pro-fibrotic response in the multifactorial disease pathogenesis. Further, treatment with the anti-fibrotic drug pirfenidone reduced pulmonary fibrosis in this model. This model may therefore aid studies of the pathogenesis and therapy of IPF.

---

#A full list of authors and their affiliations appears at the end of the paper.

Idiopathic pulmonary fibrosis (IPF) is a severe progressive lung disease with limited therapeutic options. IPF mostly affects the elderly and is characterized by altered cellular composition and aberrant fibrotic remodeling of the distal lung, leading to progressive loss of lung function and ultimately death due to respiratory failure[1–5]. Emerging evidence from genetic studies suggests that excessive production of the airway mucin MUC5B in lung epithelial cells associated with epithelial remodeling and impaired mucociliary clearance in peripheral airways may be an important trigger of IPF[6–10]. However, the role of epithelial dysfunction in the in vivo pathogenesis remains poorly understood and animal models of progressive pulmonary fibrosis have been lacking[11]. The E3 ubiquitin-protein ligase NEDD4-2 participates in diverse cellular processes that are involved in epithelial homeostasis and may be implicated in the development of chronic lung disease and fibrosis when dysregulated. First, NEDD4-2 regulates the epithelial $Na^+$ channel (ENaC) by ubiquitination, retrieval from the apical cell surface and targeting for degradation[12–15]. In the lung, ENaC is a key player implicated in the regulation of the thin film of liquid that covers airway surfaces and is essential for proper mucociliary clearance of inhaled irritants and pathogens[16,17]. Constitutive deletion of Nedd4-2 has been shown to increase surface expression and activity of ENaC in alveolar type 2 (AT2) cells of neonatal mice[18,19], and previous studies in mice overexpressing βENaC in the conducting airways (Scnn1b-Tg) demonstrated that increased ENaC activity causes enhanced transepithelial absorption of salt and water leading to airway surface liquid depletion and impaired mucociliary clearance[20,21]. Second, Nedd4-2 directs post-ER/Golgi trafficking of the 21 kDa surfactant protein-C proprotein (proSP-C) to distal compartments in alveolar epithelial cells[22]. Mutations in the SFTPC gene have been linked to interstitial lung disease in patients with familial and rare cases of sporadic IPF through a toxic gain of function mechanism[23]. Third, Nedd4-2 terminates the transforming growth factor β (TGFβ) induced signal transduction by ubiquitination of linker phosphorylated active Smad2/3[24]. TGFβ is a known potent inducer of organ fibrosis and is also found in increased levels in IPF. Inhibition of the TGFβ signaling pathway is able to prevent fibrotic changes in rodent models of pulmonary fibrosis[25]. Constitutive deletion of Nedd4-2 in lung epithelial cells in mice was shown to cause neonatal lethality due to severe lung disease with massive pulmonary inflammation resulting in premature death 2–3 weeks after birth[18,19]. Here, we generate mice with conditional deletion of Nedd4-2 in lung epithelial cells at adult ages and determined effects on lung morphology and function. We demonstrate a progressive evolution of a clinical, radiological, physiological, and histological phenotype that overlaps with those found in patients with IPF. Using this model, we investigate the impact of ENaC dysregulation on airway surface liquid and mucociliary clearance, alterations in surfactant component expression, and endogenously augmented TGFβ responses as initiating and perpetuating factors in the pathogenesis of IPF-like disease. Further, we determine NEDD4-2 expression in lung tissue of IPF patients and compare changes in the lung proteome of mice with conditional deletion of Nedd4-2 and patients with IPF to define a common fibrotic signature and biological pathways involved in the pathogenesis of IPF. Finally, we use interventional therapy with pirfenidone in this mouse model to underscore its eligibility as a preclinical model and a platform for discovery of new therapeutic strategies for IPF.

## Results

**NEDD4-2 expression is reduced in lungs from patients with IPF.** Previous transcriptome analyses showed reduced levels of NEDD4-2 transcripts in lung tissues from IPF patients[26,27], however, alterations in NEDD4-2 protein expression in IPF have not been reported. We therefore compared NEDD4-2 protein and transcript expression between lung tissue biopsies from IPF patients and age-matched controls (Fig. 1, Supplementary Table 1). Immunohistochemistry demonstrated reduced NEDD4-2 expression in epithelial cells lining the distal airways of IPF patients compared to controls (Fig. 1a). These findings were corroborated by quantitative mass spectrometry showing a substantial reduction of NEDD4-2 levels (~60%) in IPF patients compared to controls (Fig. 1b). A similar reduction was observed for NEDD4-2 transcripts (Fig. 1c) confirming previous transcriptome analyses in independent IPF patient cohorts[26,27].

**Conditional deletion of Nedd4-2 causes pulmonary fibrosis.** To determine effects of conditional deletion of Nedd4-2 in epithelial cells of the adult mouse lung, we crossed mice carrying Nedd4-2 flanked by loxP sites (Nedd4-2^{fl/fl}) with CCSP-rtTA2^S-M2/LC-1 mice enabling tight doxycycline-induced rtTA-mediated expression of Cre recombinase producing deletion of Nedd4-2 in AT2 cells and club cells of the conducting airways[18,28,29] (Supplementary Fig. 1a). Adult triple transgenic Nedd4-2^{fl/fl}/CCSP-rtTA2^S-M2/LC1 mice, hereafter referred to as conditional Nedd4-2^{−/−} mice, and littermate controls were induced with doxycycline for up to 4 months. After approximately 3 months on doxycycline, we observed spontaneous mortality associated with severe weight loss and hypoxia resulting in an overall mortality of 70% at 4 months in the conditional Nedd4-2^{−/−} mice (Fig. 2a–c). Pulmonary function testing showed progressive restriction with a 45% decrease in static compliance at 4 months (Fig. 2d). Lung imaging using micro-computed tomography (micro-CT) demonstrated morphological hallmarks of IPF[1] including reticular opacities, traction bronchiectasis, and honeycombing-like

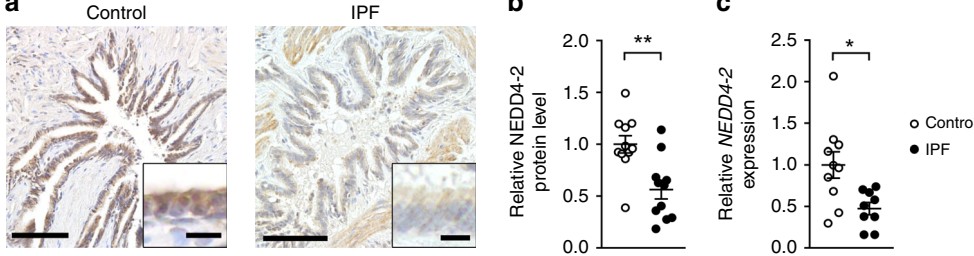

**Fig. 1 NEDD4-2 expression is reduced in lung tissue biopsies from patients with IPF. a** Micrographs of lung sections from patients with IPF and age-matched controls stained with anti-NEDD4-2 antibody (representative of $n = 9$/group). Scale bars, 100 and 10 μm (insets). **b, c** Levels of NEDD4-2 protein as determined by parallel reaction monitoring mass spectrometry ($n = 11$/group) (**b**) and NEDD4-2 mRNA (control, $n = 10$; IPF, $n = 9$) (**c**) in lung biopsies from IPF patients and controls. *$P < 0.05$, **$P < 0.01$ compared to controls. Statistical analysis was performed with unpaired two-tailed $t$ test. Data are shown as mean ± S.E.M. Source data are provided in the Source Data file.

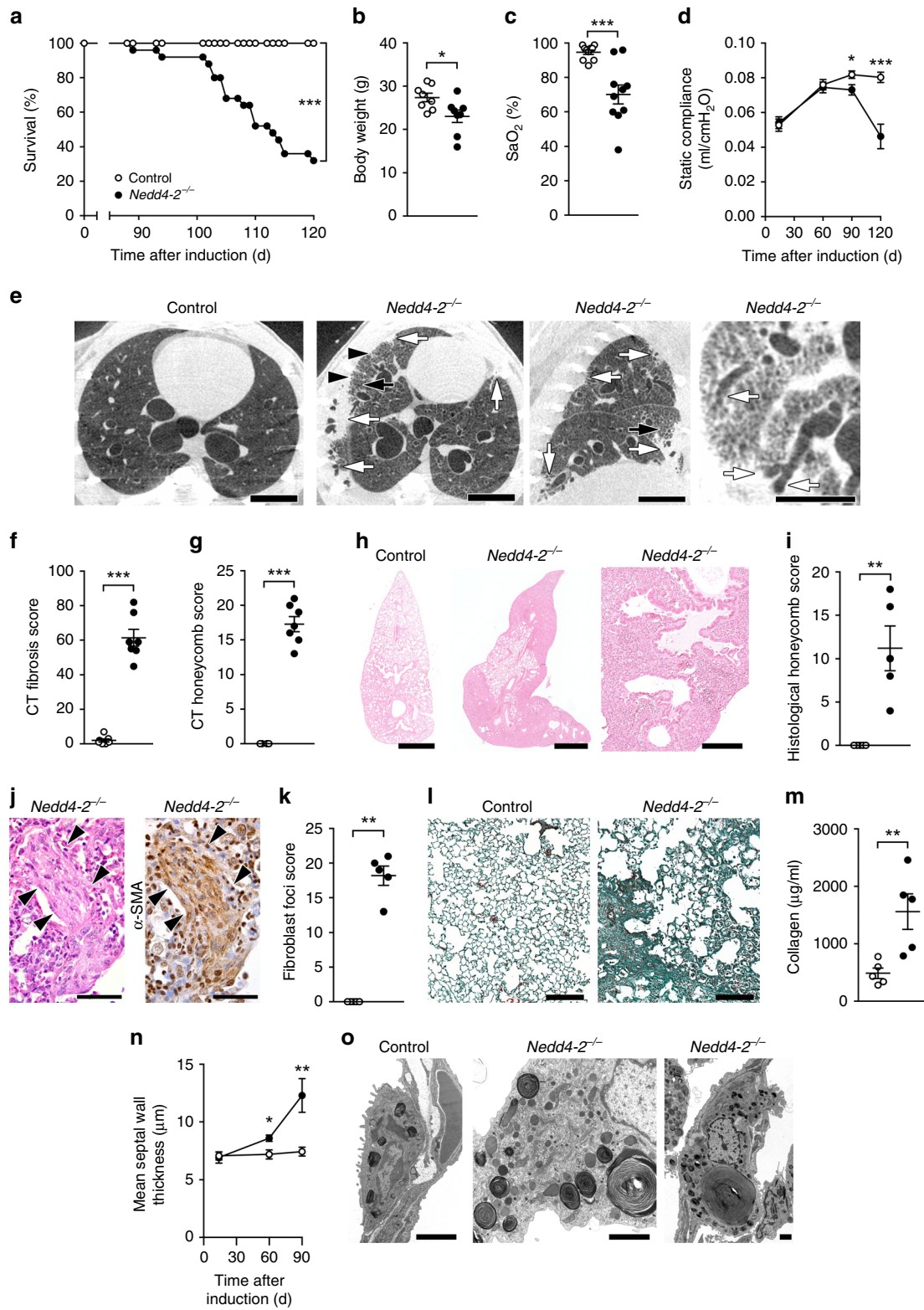

cysts with spatially heterogeneous distribution in peripheral regions of the lung (Fig. 2e). The CT scores for fibrosis and honeycombing-like cysts were increased in conditional *Nedd4-2*−/− mice compared to controls (Fig. 2f, g). Microscopic morphological analysis of lung sections revealed areas of patchy fibrosis of the alveolar interstitium associated with inflammatory infiltrates, α smooth muscle actin (αSMA) positive fibroblast foci-like structures, increased collagen deposition, septal wall

thickening, hypertrophy of AT2 cells, subpleural microscopic honeycombing-like cysts, and Muc5b-positive material within fibrotic areas of the peripheral airspaces (Fig. 2h–n and see below). By transmission electron microscopy we found ultra-structural abnormalities in $15.0 \pm 4.5\%$ of all AT2 cells in lung regions that appeared macroscopically normal that were increased to $56.1 \pm 9.9\%$ of all AT2 cells in fibrotic regions of conditional *Nedd4-2*−/− mice (mean $\pm$ SEM, $n = 3$; $P < 0.05$,

**Fig. 2 Conditional deletion of *Nedd4-2* in adult lung epithelial cells causes pulmonary fibrosis. a** Survival curves of conditional *Nedd4-2*[−/−] mice and littermate controls ($n = 26$/group). **b, c** Summary of body weight ($n = 8$/group) (**b**) and oxygen saturation ($n = 10$/group) (**c**) of conditional *Nedd4-2*[−/−] and control mice induced with doxycycline for 3–4 month recorded at day of endpoint study. **d** Lung compliance was assessed by pulmonary function testing after 0.5 (control, $n = 6$ mice; conditional *Nedd4-2*[−/−], $n = 5$ mice), 2 (control, $n = 5$ mice; conditional *Nedd4-2*[−/−], $n = 7$ mice), 3 (control, $n = 10$ mice; conditional *Nedd4-2*[−/−], $n = 13$ mice), and 4 months (control, $n = 9$ mice; conditional *Nedd4-2*[−/−], $n = 6$ mice) of doxycycline induction. **e–g** Micro-CT imaging studies were performed after an average of 4 month of doxycycline induction when conditional *Nedd4-2*[−/−] mice developed clinical symptoms ($n = 7$/group). **e** Representative micro-CT images showing subpleural patchy consolidation with traction bronchiectasis (white arrows), subpleural cystic honeycombing-like destruction of lung parenchyma adjacent to consolidation (black arrows) and subpleural reticulation with parenchymal lines (black arrowheads) in conditional *Nedd4-2*[−/−] mice. Scale bars, 3 mm. **f, g** Quantification of pulmonary fibrosis (**f**) and honeycombing-like cysts (**g**) by micro-CT scoring. **h–l** Histomorphological evaluation of lungs of clinically symptomatic conditional *Nedd4-2*[−/−] mice after an average of 4 months of doxycycline induction ($n = 5$/group). **h, i** Micrographs of representative hematoxylin and eosin (H&E) stained lung sections. Scale bars, 1 mm (low magnification overview) and 200 μm (high magnification). **i** Quantification of honeycombing-like cysts determined from H&E stained lung sections. **j** Micrograph of representative adjacent lung sections stained with H&E and anti-α-smooth muscle actin (αSMA) antibody depicting fibroblast foci-like alterations (arrowheads). Scale bars, 50 μm. **k** Quantification of fibroblast foci-like alterations as determined from H&E stained lung sections. **l** Masson–Goldner–Trichrome staining of lung sections depicts increased collagen deposition in the lung parenchyma of conditional *Nedd4-2*[−/−] mice. Scale bars, 200 μm. **m** Collagen content (Sircol) in lungs of conditional *Nedd4-2*[−/−] mice and littermate controls ($n = 5$/group). **n** Mean septal wall thickness in conditional *Nedd4-2*[−/−] mice and littermate controls after 0.5, 2, and 3 months of doxycycline induction ($n = 6$/group). **o** Transmission electron microscopy reveals ultrastructural abnormalities in alveolar type 2 cells in conditional *Nedd4-2*[−/−] mice ($n = 6$/group). Scale bars, 2 μm. \*$P < 0.05$, \*\*$P < 0.01$, \*\*\*$P < 0.001$ compared to littermate controls of the same age. Survival in **a** was analyzed with log rank test. Statistical analysis was performed with unpaired two-tailed $t$ test in (**b, d, n**), and with two-tailed Mann–Whitney test in (**c, f, g, i, k, m**). Data are shown as mean ± S.E.M. Source data are provided in the Source Data file.

unpaired two-tailed $t$ test) (Fig. 2o). Longitudinal assessment of inflammatory cells and pro-inflammatory cytokines in bronchoalveolar lavage (BAL) fluid demonstrated that macrophages were morphologically activated and total cells, macrophages, neutrophils and cytokines associated with IPF[30] such as interleukin-1β (IL-1β), keratinocyte-derived chemokine (KC), and interleukin-13 (IL-13) were increased from 2 to 3 months onward in conditional *Nedd4-2*[−/−] compared to control mice (Supplementary Fig. 2). Control experiments in mice carrying different combinations of transgenes showed that progressive pulmonary fibrosis was only observed after conditional deletion of *Nedd4-2* and could not be explained by expression of rtTA or Cre recombinase, nor by application of doxycycline alone (Supplementary Fig. 1b–e).

**Airway remodeling and increased production of Muc5b.** Because recent studies have identified epithelial remodeling with bronchiolization and ectopic expression of the secreted mucin MUC5B in peripheral airways and honeycomb cysts as key features of IPF[6,8–10], we next evaluated cell type composition, i.e., numeric densities of CCSP-positive club cells, tubulin-positive ciliated cells and Alcian blue-periodic acid–Schiff (AB-PAS)-positive goblet cells, as well as epithelial expression of Muc5b and Muc5ac, along the conducting airways (Fig. 3a–i). These studies showed epithelial remodeling of the peripheral airways characterized by a decrease in club cells and increase in ciliated cells and goblet cells in distal and terminal airways of conditional *Nedd4-2*[−/−] compared to control mice (Fig. 3e–g). These abnormalities were associated with elevated transcript levels of *Muc5b* and *Muc5ac* in lungs of conditional *Nedd4-2*[−/−] mice (Fig. 3j, k). Immunolocalization studies detected elevated expression of Muc5b in epithelial cells along the tracheobronchial tree extending into terminal airways, whereas increased expression of Muc5ac was restricted to proximal and distal airways of conditional *Nedd4-2*[−/−] mice (Fig. 3h, i). Further, Muc5b immunoreactive signals were detected in honeycomb-like cysts observed in conditional *Nedd4-2*[−/−] mice (Fig. 3c). In addition to *Muc5b*, we evaluated transcript levels of a panel of 12 genes recently identified as risk factors for pulmonary fibrosis in genome wide association studies in patients[7,31–33]. We found that eight of these genes including *Muc5b*, *Atp11a*, *Dpp9*, *Disp2*, *Wnt9b*, *Tert*, *Terc*, and *Fam13a*, implicated in a spectrum of homeostatic functions such as host defense, barrier function and

telomere maintenance, were differentially expressed in lungs of conditional *Nedd4-2*[−/−] mice compared to controls (Fig. 3, Supplementary Fig. 3).

**Reduced airway surface liquid and mucociliary transport.** Recent evidence suggests that impaired mucociliary clearance caused by overexpression of Muc5b contributes to the pathogenesis of IPF[6,34]. Because increased ENaC function has also been implicated in the pathogenesis of airway surface dehydration/mucus hyperconcentration and mucociliary dysfunction[16,17], we investigated the effects of conditional deletion of *Nedd4-2* on ENaC activity, airway surface liquid volume regulation and mucociliary clearance in our model (Fig. 4). Transepithelial bioelectric measurements performed after 2 weeks of doxycycline induction demonstrated increased (approximately threefold) amiloride-sensitive ENaC-mediated Na[+] currents in freshly excised airway tissues of conditional *Nedd4-2*[−/−] compared to control mice (Fig. 4a). This increase in ENaC activity was associated with reduced airway surface liquid height (Fig. 4b, c) and reduced mucociliary transport (MCT) velocity (Fig. 4d, e) on primary airway cultures of conditional *Nedd4-2*[−/−] mice compared to controls.

**proSP-C mistrafficking does not contribute to lung fibrosis.** Nedd4-2 has also been shown to be involved in the posttranslational regulation of SP-C biosynthesis[22]. In immunofluorescence studies we found markedly altered intracellular distribution of proSP-C in AT2 cells isolated from conditional *Nedd4-2*[−/−] mice compared to controls (Fig. 5a). Further, the banding pattern of proSP-C isoforms in Western blots demonstrated a shift to the unprocessed isoforms resulting in decreased amounts of mature SP-C in BAL of conditional *Nedd4-2*[−/−] mice (Fig. 5b–d). Despite these effects of Nedd4-2 deficiency on proSP-C post-translational processing, other components of the surfactant system were largely unaffected (Fig. 5b). To elucidate the specific contribution of proSP-C mistrafficking to the fibrotic lung phenotype in the setting of Nedd4-2 deficiency, we crossed conditional *Nedd4-2*[−/−] mice to *Sftpc*-deficient (*Sftpc*[−/−]) mice. In these studies, we did neither observe an amelioration nor aggravation of the lung disease phenotype when *Sftpc* was deleted in conditional *Nedd4-2*[−/−] mice suggesting that multiple pathways link Nedd4-2 deficiency to pulmonary fibrosis (Fig. 5e–g).

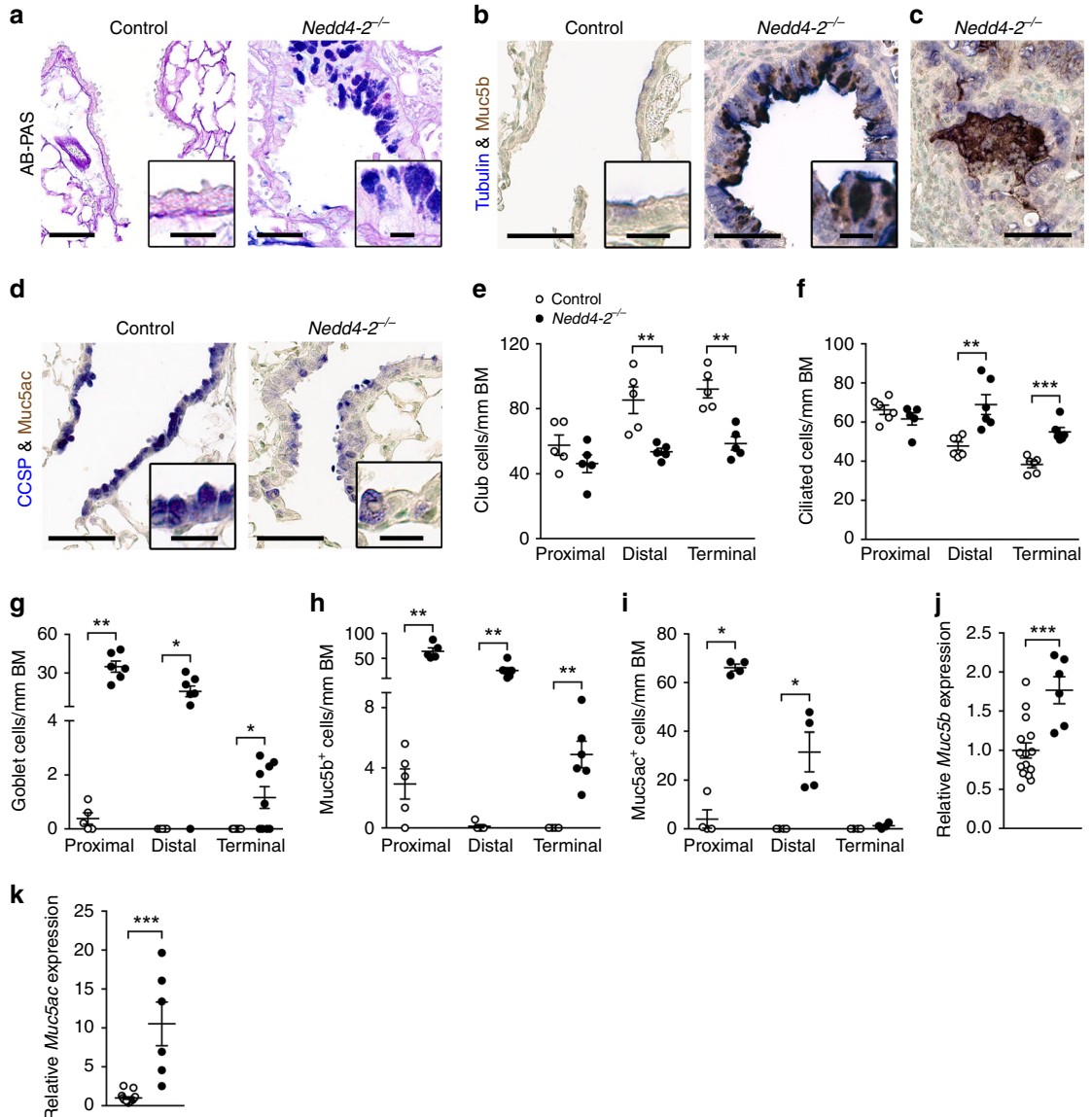

**Fig. 3 Conditional deletion of *Nedd4-2* in lung epithelial cells causes peripheral airway remodeling with increased Muc5b production. a–d** Representative micrographs and high magnification insets illustrating the airway cell population of terminal bronchioles from conditional *Nedd4-2⁻/⁻* mice and littermate controls after 3 months of doxycycline induction stained with Alcian blue and periodic acid-Schiff (AB-PAS) (control, $n = 5$ mice; *Nedd4-2⁻/⁻*, $n = 9$ mice) (**a**), or co-stained with anti-Muc5b and anti-acetylated α-tubulin antibodies (control, $n = 5$ mice; *Nedd4-2⁻/⁻*, $n = 6$) (**b, c**), or with anti-Muc5ac and anti-CCSP antibodies ($n = 4$/group) (**d**). Scale bars, 60 and 15 μm (insets). **e–i** Summary of numeric densities of club cells ($n = 5$ mice/group) (**e**), ciliated cells ($n = 6$/group and proximal, conditional *Nedd4-2⁻/⁻*, $n = 5$ mice) (**f**), goblet cells (proximal, control, $n = 5$ mice; conditional *Nedd4-2⁻/⁻*, $n = 6$ mice; distal, control, $n = 5$ mice; conditional *Nedd4-2⁻/⁻*, $n = 7$ mice, terminal, $n = 9$/group) (**g**), Muc5b expressing cells (proximal, $n = 5$/group; distal, control, $n = 5$ mice; conditional *Nedd4-2⁻/⁻*, $n = 6$ mice, terminal, $n = 6$/group) (**h**), and Muc5ac expressing cells ($n = 4$/group) (**i**) in proximal, distal and terminal conducting airways of conditional *Nedd4-2⁻/⁻* mice and littermate controls after 3 months of doxycycline induction (BM basement membrane). **j, k** mRNA expression levels of *Muc5b* (**j**) and *Muc5ac* (control, $n = 15$ mice; conditional *Nedd4-2⁻/⁻*, $n = 6$ mice) (**k**) in whole lungs from 3-month induced conditional *Nedd4-2⁻/⁻* and control mice. $*P < 0.05$, $**P < 0.01$, $***P < 0.001$. Statistical analysis in **e, g, h, i, k** was performed with two-tailed Mann–Whitney test, and in **f**, and **j** with unpaired two-tailed *t* test. Data are shown as mean ± S.E.M. Source data are provided in the Source Data file.

**Increased levels of TGFβ and exaggerated Smad2/3 signaling.** In addition to the regulation of ENaC and proSP-C, Nedd4-2 has been implicated in the regulation of pro-fibrotic TGFβ signaling by polyubiquitination and subsequent degradation of phosphorylated Smad2/3, the main intracellular mediators of the TGFβ signaling pathway[24]. We found that the onset and progression of pulmonary fibrosis was associated with elevated levels of active TGFβ in lungs of conditional *Nedd4-2⁻/⁻* mice (Fig. 6a). Consistent with fibrotic remodeling, transcript levels of the pro-

fibrotic markers vimentin (*Vim*) and fibronectin (*Fn1*) were increased, while transcripts of AT2 cell markers such as surfactant protein C (*Sftpc*) and surfactant protein D (*Sftpd*) were decreased in lungs of conditional *Nedd4-2⁻/⁻* mice (Fig. 6b, c). Stimulation of primary AT2 cells with TGFβ resulted in highly elevated levels of phosphorylated Smad2/3 in cells from conditional *Nedd4-2⁻/⁻* compared to control mice (Fig. 6d–f). A similar pattern was observed for downstream target genes of TGFβ signaling such as *Serpine1*, *Smad7*, and Ski-like (*Skil*) (Fig. 6g–i).

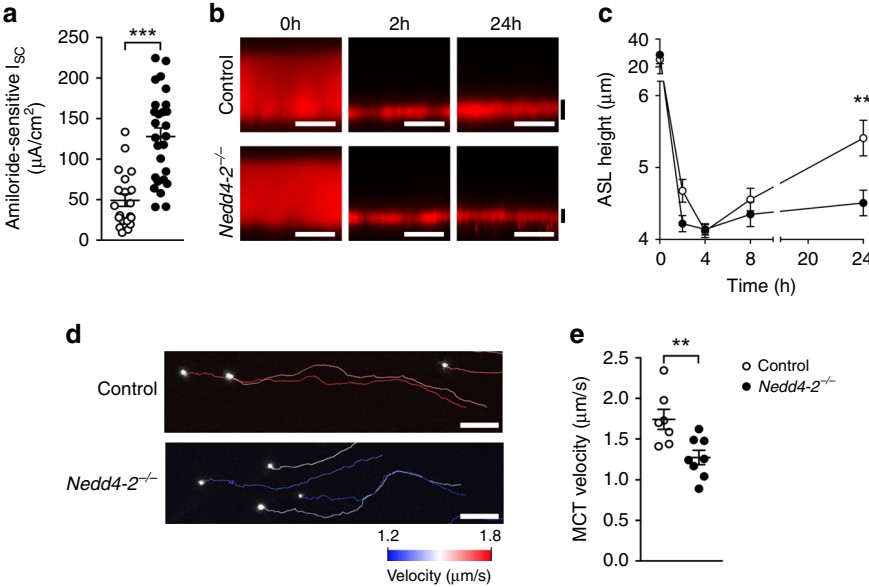

**Fig. 4 Conditional deletion of *Nedd4-2* in airway epithelial cells causes increased ENaC activity, reduced airway surface liquid (ASL) height and impaired mucociliary transport. a** Amiloride-sensitive short circuit current ($I_{sc}$) in freshly excised airway tissues of conditional *Nedd4-2*$^{-/-}$ and control mice (control, $n = 20$ mice; conditional *Nedd4-2*$^{-/-}$, $n = 27$ mice) induced with doxycycline for 2 weeks. **b, c** Confocal images of ASL height (**b**) and summary of measurements (**c**) at baseline and 2, 4, 8, and 24 h after application of rhodamine dextran to primary murine airway epithelial cultures from conditional *Nedd4-2*$^{-/-}$ mice and littermate controls induced with doxycycline for 2 weeks ($n = 3$ independent experiments performed in triplicates). Scale bars, 15 μm. **d, e** Mucociliary transport (MCT) velocity was determined from transport rates of fluorescent beads added on the surface of primary murine airway epithelial cultures. Representative bead tracks that visualize MCT velocity (**d**) and summary of measurements (**e**) in primary cultures from conditional *Nedd4-2*$^{-/-}$ mice and littermate controls ($n = 3$ independent experiments). Scale bars, 50 μm. **P < 0.01, ***P < 0.001. Statistical analysis in **a** was performed with two-tailed Mann–Whitney test and in **c, e** with unpaired two-tailed *t* test. Data are shown as mean ± S.E.M. Source data are provided in the Source Data file.

**Lung proteome of conditional *Nedd4-2*$^{-/-}$ mice and IPF patients.** Next, we performed proteome profiling by mass spectrometry to characterize molecular changes associated with pulmonary fibrosis in conditional *Nedd4-2*$^{-/-}$ mice at the whole organ level and compared these abnormalities to patients with IPF (Supplementary Table 1). Overall, we quantified 4539 proteins from mouse lung and 2834 proteins from human lung tissues. Totally, 547 proteins were differentially expressed in lungs of conditional *Nedd4-2*$^{-/-}$ mice ($P < 0.05$) and 322 proteins were changed in lung tissues of IPF patients compared to their respective controls ($P < 0.05$) with a substantial overlap of 104 differentially detected proteins between conditional *Nedd4-2*$^{-/-}$ mice and patients with IPF (Fig. 7a–c). Clustering of matrisome-annotated proteins[35] identified substantial alterations in extracellular matrix (ECM) components in conditional *Nedd4-2*$^{-/-}$ mice and IPF patients (Fig. 7a, b). Of the 104 mutually changed proteins, 16 proteins were assigned to the matrisome (Fig. 7c, d). These included structural ECM and related proteins such as collagen 14a1 (COL14A1), tenascin (TNC) and SERPINH1, which were commonly up-regulated in the lungs of conditional *Nedd4-2*$^{-/-}$ mice and IPF patients (Fig. 7d). Upregulation of these pro-fibrotic proteins was confirmed by immunohistochemistry of lung sections from conditional *Nedd4-2*$^{-/-}$ mice and IPF patients, and their respective controls (Fig. 7e). In addition, we observed that basal lamina proteins collagen 4a1 (COL4A1), collagen 4a2 (COL4A2), and collagen 4a3 (COL4A3) were downregulated, exemplifying abnormalities of lung parenchymal architecture at the molecular level (Fig. 7d).

**Pirfenidone reduces fibrosis in conditional *Nedd4-2*$^{-/-}$ mice.** Finally, we treated conditional *Nedd4-2*$^{-/-}$ mice with pirfenidone, an approved drug for the treatment of IPF[36,37], and determined effects on pulmonary fibrosis and inflammation.

Treatment was initiated two months after doxycycline induction, i.e., when lungs of conditional *Nedd4-2*$^{-/-}$ mice showed first signs of fibrotic remodeling (Fig. 2n), and was continued for four weeks. Conditional *Nedd4-2*$^{-/-}$ mice treated orally with pirfenidone showed a reduced volume of fibrotic lesions detected by micro-CT, improved static compliance in pulmonary function testing and decreased concentrations of active TGFβ in lung homogenates (Fig. 8a–d). Further, inflammatory markers including neutrophils and levels of IL-13 and IL-1β in BAL were reduced in lungs of pirfenidone-treated conditional *Nedd4-2*$^{-/-}$ mice (Fig. 8e–g). At the cellular level, when primary AT2 cells were pre-treated with pirfenidone in vitro (Fig. 8h–k), we observed a reversal of their hyperresponsiveness to TGFβ (Fig. 6d–f). Levels of phosphorylated Smad2 in these cells from conditional *Nedd4-2*$^{-/-}$ mice pretreated with pirfenidone were in the range of control mice (Fig. 8h). A similar pattern was observed for downstream target genes of TGFβ signaling such as *Serpine1*, *Smad7*, and *Skil* (Fig. 8i–k).

**Pirfenidone treatment reverts changes in the lung proteome.** To determine the pathways and biological functions altered by pirfenidone treatment in our model, lung tissues of pirfenidone-treated and untreated *Nedd4-2*$^{-/-}$ mice, and respective littermate controls were analyzed by mass spectrometry (Fig. 9). In total, 4565 proteins were quantified and 705 proteins were differentially regulated among the four experimental groups. Totally, 54 of these proteins could be assigned to the matrisome (Fig. 9a). Similar to the independent experiments summarized in Fig. 7a, we observed a clear separation in proteome profiles between untreated *Nedd4-2*$^{-/-}$ mice and controls (Fig. 9a). Of note, pirfenidone treatment reverted the proteomic changes associated with pulmonary fibrosis in conditional *Nedd4-2*$^{-/-}$ mice to a more normal lung proteome of untreated control mice (Fig. 9a, b).

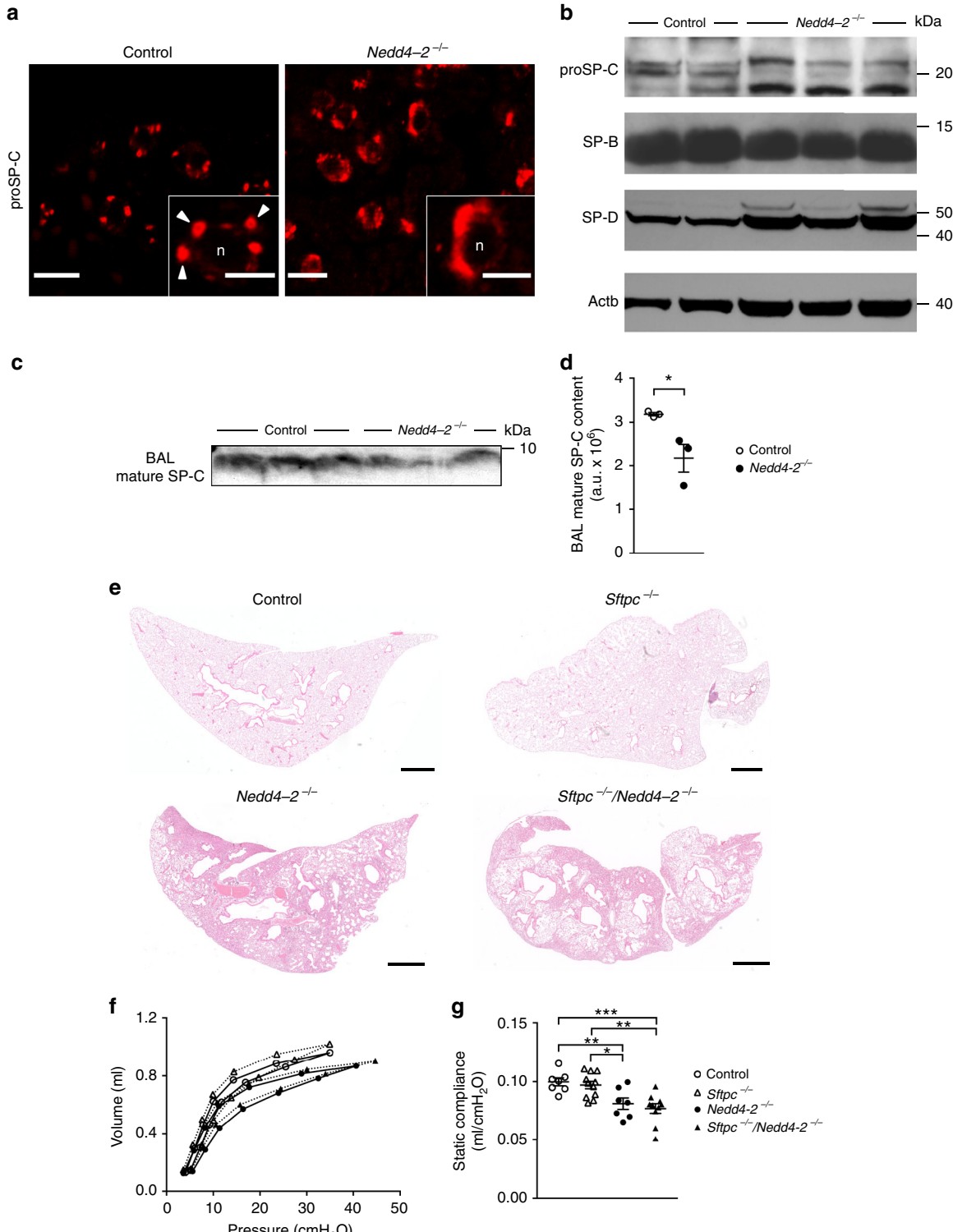

Correlations of fold changes derived from the respective comparisons confirmed this observation for regulated matrisome assigned proteins, as well as all regulated proteins (Supplementary Fig. 5). All significantly changed proteins ($P < 0.01$) derived from the two key comparisons of conditional $Nedd4$-$2^{-/-}$ vs. control mice ($n = 565$) and conditional $Nedd4$-$2^{-/-}$ vs. pirfenidone-treated $Nedd4$-$2^{-/-}$ mice ($n = 384$) were analyzed using Ingenuity Pathway Analysis to identify processes and pathways affected by pirfenidone treatment (Fig. 9c, d). These analyses showed that pirfenidone treatment restored expression of downregulated

proteins assigned to cell death mechanisms and suppressed activation of pro-inflammatory processes such as leukocyte recruitment in conditional $Nedd4$-$2^{-/-}$ mice (Fig. 9c). Furthermore, pirfenidone had a normalizing effect on canonical pathways predicted to be downregulated in conditional $Nedd4$-$2^{-/-}$ mice, such as paxillin, ERK/MAPK, and PAK pathways, and "agrin interactions" (Fig. 9c). In line with this, pirfenidone treatment counteracted the activation of the PTEN signaling pathway in lungs of conditional $Nedd4$-$2^{-/-}$ mice (Fig. 9d). Gene set enrichment analysis showed that most of the regulated proteins were

**Fig. 5 Lack of *Nedd4-2* causes misprocessing of proSP-C in alveolar type 2 (AT2) cells, but genetic deletion of Sftpc does not prevent pulmonary fibrosis in conditional *Nedd4-2⁻/⁻* mice. a** Representative immunofluorescence images of lungs from conditional *Nedd4-2⁻/⁻* mice and littermate controls stained with anti-proSP-C antibodies. The subcellular distribution of proSP-C in control lungs represents predominantly large subplasma membrane organelles consistent with lamellar bodies (white arrowheads) while expression of proSP-C in conditional *Nedd4-2⁻/⁻* lungs also occurs in smaller cytosolic vesicles ($n = 3$/group). Scale bars, 50 and 10 μm (insets). **b** Western blot of isolated AT2 cells demonstrating aberrations in the proSP-C processing profile in conditional *Nedd4-2⁻/⁻* compared to control mice ($n = 3$/group). **c, d** Representative Western blot (**c**) and densitometric analysis (**d**) of mature SP-C in bronchoalveolar lavage fluid from conditional *Nedd4-2⁻/⁻* mice and littermate controls ($n = 3$/group). **e–g** Micrographs of representative hematoxylin and eosin (H&E) stained lung sections (scale bars, 1 mm) (**e**) and measurements of pressure–volume curves ($n = 13$/group) (**f**) and static lung compliance (**g**) from conditional *Nedd4-2⁻/⁻* ($n = 7$ mice), conditional *Nedd4-2⁻/⁻/Sftpc⁻/⁻* ($n = 10$ mice), littermate *Sftpc⁻/⁻* ($n = 11$ mice), and control mice ($n = 7$ mice) induced with doxycycline for an average of 4 months. *$P < 0.05$, **$P < 0.01$, ***$P < 0.001$. Statistical analysis was performed with unpaired two-tailed $t$ test in **d** and ANOVA with Tukey's post hoc test in **g**. Data are shown as mean ± S.E.M. Source data are provided in the Source Data file.

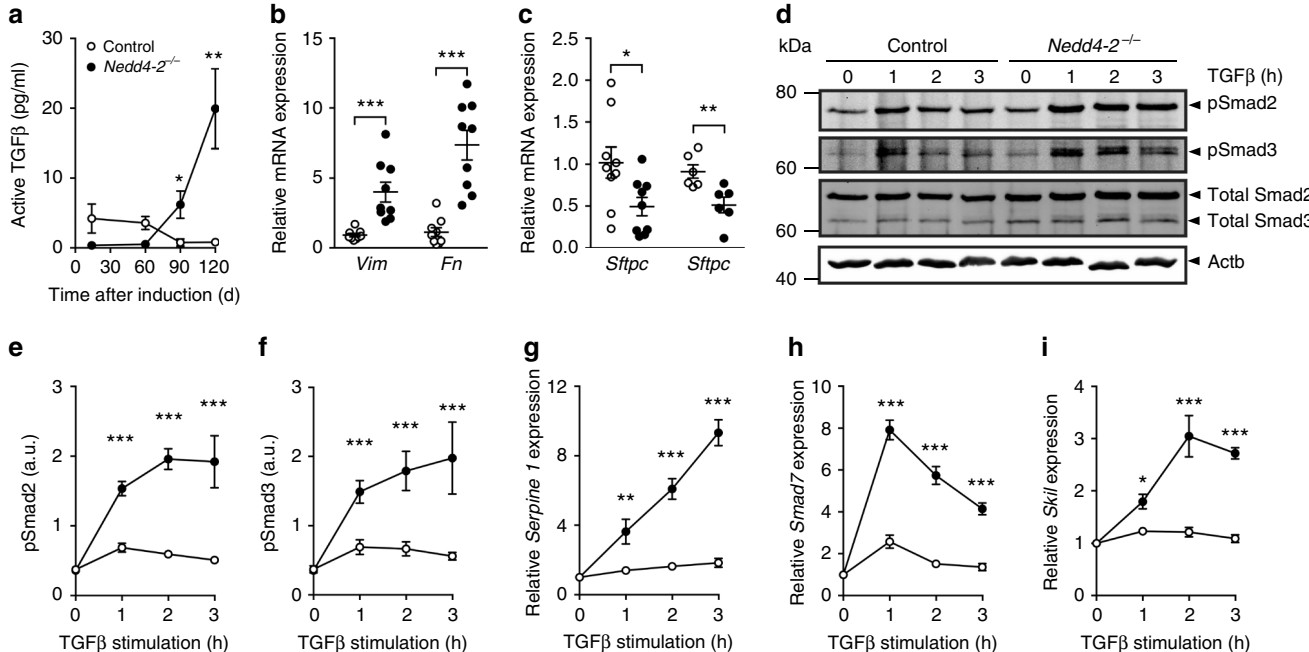

**Fig. 6 Conditional deletion of *Nedd4-2* in lung epithelial cells causes increased levels of TGFβ and exaggerated Smad2/3 signaling. a** Levels of active TGFβ in whole lung of conditional *Nedd4-2⁻/⁻* mice and littermate controls after 0.5 ($n = 4$/group), 2 ($n = 4$/group), 3 ($n = 5$/group), and 4 months ($n = 6$/group) of doxycycline induction. **b, c** mRNA expression levels of the mesenchymal marker genes Vim and Fn1 ($n = 9$/group) (**b**) and alveolar type 2 (AT2) cell markers Sftpc ($n = 9$/group) and Sftpd ($n = 6$/group) (**c**) of whole lungs from conditional *Nedd4-2⁻/⁻* and control mice after 3 months of doxycycline induction. **d–i** AT2 cells isolated from conditional *Nedd4-2⁻/⁻* and control mice induced with doxycycline for 2 weeks were pretreated with 1 ng/ml TGFβ and analyzed for phosphorylated Smad2/3 (pSmad2/3) and transcript levels of TGFβ-inducible genes. **d–f** Representative Western blot (**d**) and densitometric analysis of pSmad2 (**e**) and pSmad3 (**f**) at baseline and after 1–3 h of TGFβ stimulation. **g–i** mRNA levels of Serpine1 (**g**), Smad7 (**h**), and Skil (**i**) after the indicated time of TGFβ stimulation. All in vitro data are derived from 3 independent experiments. *$P < 0.05$, **$P < 0.01$, ***$P < 0.001$ compared to controls. Statistical analysis was performed using two-tailed Mann–Whitney test in **a, b**, two-tailed unpaired $t$ test in **c**, and two-way ANOVA with Sidac's post hoc test in **e–i**. Data are shown as mean ± S.E.M. Source data are provided in the Source Data file.

annotated to "Gene Ontology Cellular Component" terms such as "extracellular matrix", "focal adhesions", "plasma membrane", and "intracellular vesicles" (Supplementary Fig. 6).

## Discussion

In this report, we demonstrate that NEDD4-2 is reduced in lung epithelial cells of patients with IPF (Fig. 1, Supplementary Fig. 4) and that conditional deletion of *Nedd4-2* in lung epithelial cells of adult mice results in a spontaneous, chronic progressive, restrictive lung disease that shares clinical, radiological, and histopathological key features with IPF. These features include patchy pulmonary fibrosis with honeycombing-like cysts, fibroblast foci-like structures, bronchiolization of peripheral airways, altered pulmonary expression of *Muc5b* and other IPF modifier genes, a commonly altered proteomic signature, and death due to

respiratory failure (Fig. 2, Fig. 3, and Fig. 7). This spontaneous and progressive development of pulmonary fibrosis in conditional *Nedd4-2⁻/⁻* mice is strikingly different from most other mouse models of IPF, which are based on exogenous administration of toxic substances such as bleomycin that induce acute injury and repair resulting in a transient fibrotic response that does not recapitulate the evolution of IPF in patients[11,38,39]. Of note, reduced *NEDD4-2* transcript levels were documented in previous transcriptomics studies in several independent IPF cohorts[26,27]. Collectively, our data implicate NEDD4-2 deficiency in the pathogenesis of IPF and have established a mouse model of this devastating disease.

The E3 ubiquitin-protein ligase NEDD4-2 is involved in the regulation of a number of proteins including ENaC, Smad2/3, and proSP-C that play key roles in epithelial ion and fluid transport, regulation of TGFβ signaling and surfactant biogenesis

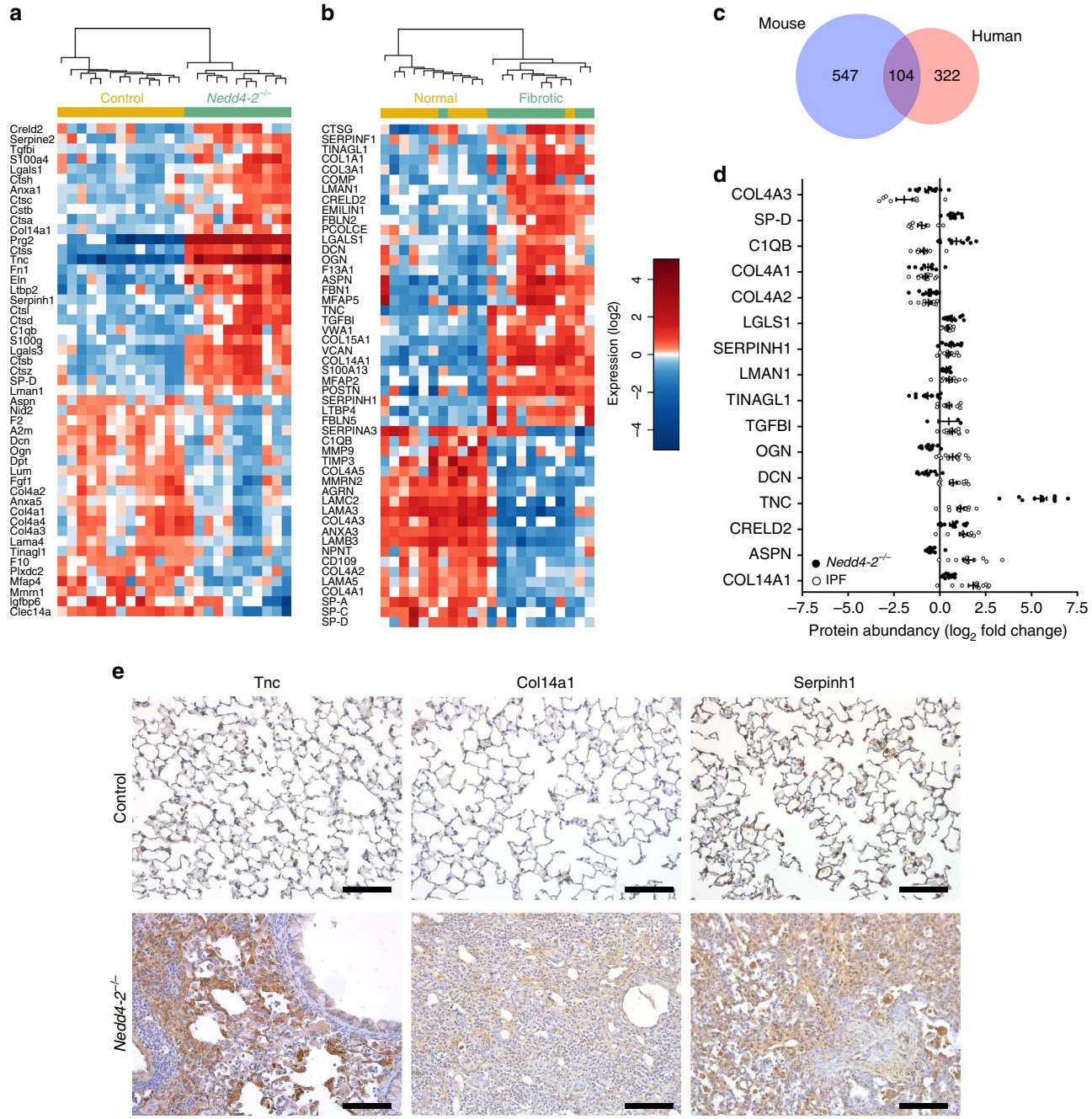

**Fig. 7 Comparison of proteomic signatures of pulmonary fibrosis in conditional *Nedd4-2*$^{-/-}$ mice and patients with IPF. a, b** Heatmap of differentially regulated matrisome annotated proteins in lungs from conditional *Nedd4-2*$^{-/-}$ mice ($n = 11$) and control mice ($n = 13$) induced with doxycycline for 3 months (**a**), and in lung tissues from IPF patients and controls ($n = 11$/group) (**b**). **c** Venn-diagram showing proportion of unique and common differentially regulated proteins in lungs of conditional *Nedd4-2*$^{-/-}$ mice and IPF patients. **d** Summary of differentially regulated matrisome proteins found in conditional *Nedd4-2*$^{-/-}$ mice and human IPF samples. **e** Micrographs of representative lung sections from conditional *Nedd4-2*$^{-/-}$ mice and littermate controls stained with anti-Tnc, anti-Col14a1, and anti-Serpinh1 antibodies ($n = 6$/group). Scale bars, 100 µm.

essential for epithelial homeostasis and lung health[12–17,20–25,40]. In this context, our study provides mechanistic insight regarding the role of Nedd4-2 deficiency in the pathogenesis of IPF-like lung disease. First, our data show that conditional deletion of *Nedd4-2* in airway epithelial cells produces increased ENaC activity leading to increased transepithelial sodium/fluid absorption and depletion of airway surface liquid from airway surfaces. As previously shown in cystic fibrosis airways[16,17,20,41], increased ENaC activity and airway surface liquid depletion resulted in reduced MCT rates on primary airway cultures of conditional

*Nedd4-2*$^{-/-}$ mice (Fig. 4). Because mucociliary clearance is an important defense mechanism of the airways and its failure results in chronic epithelial injury and inflammation, these data support mucociliary dysfunction as an important extracellular pathogenic process that triggers airway remodeling and pulmonary fibrosis in conditional *Nedd4-2*$^{-/-}$ mice, as recently suggested in an independent model where mucociliary clearance was reduced due to transgenic overexpression of Muc5b and shown to be associated with more severe and persistent bleomycin-induced pulmonary fibrosis in mice[6,34]. Second, we

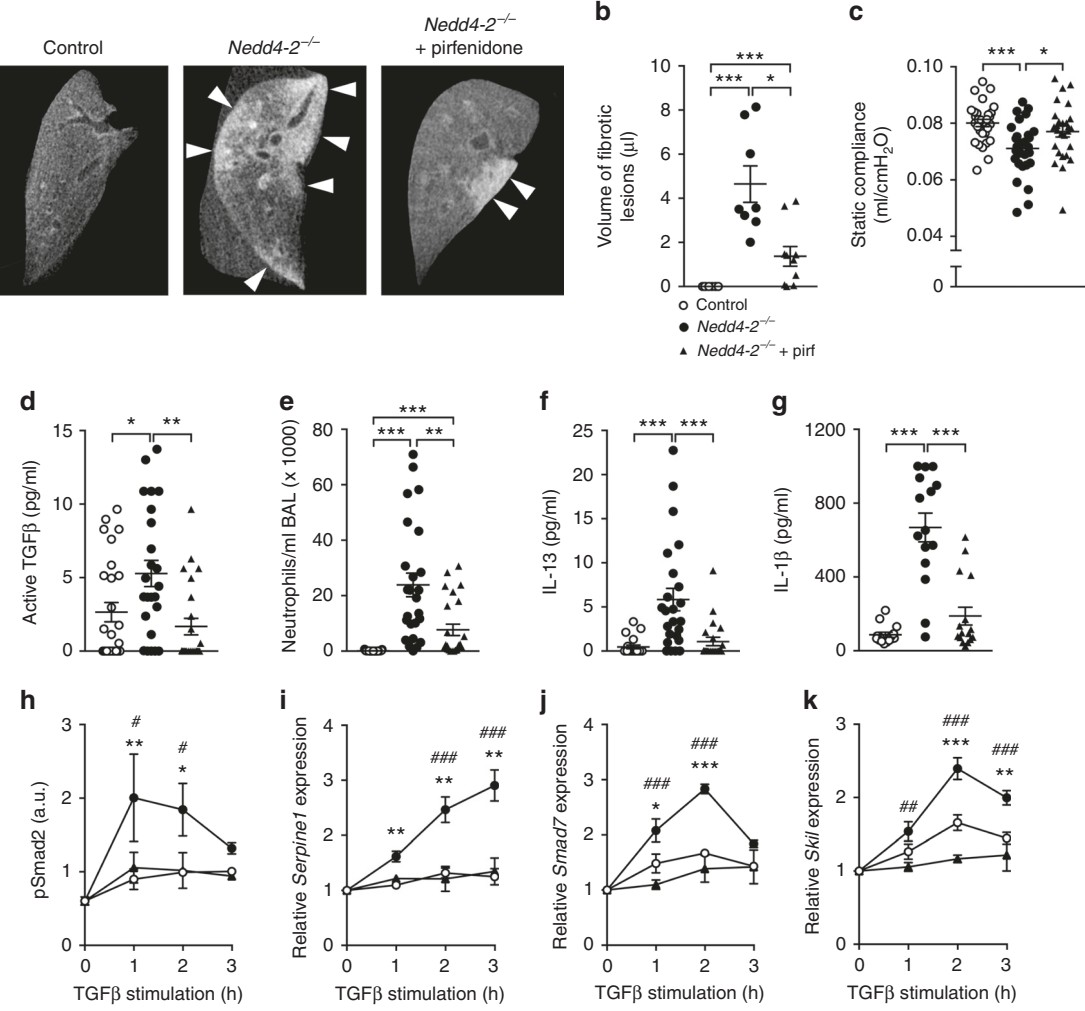

**Fig. 8 Pirfenidone treatment reduces pulmonary fibrosis and inflammation in conditional *Nedd4-2⁻/⁻* mice. a** Representative micro-CT images of paraffin embedded lungs of 3-months doxycycline-induced conditional *Nedd4-2⁻/⁻* and control mice treated with pirfenidone (pirf) or vehicle alone for one month. Fibrotic areas show patches of increased tissue density (white arrowheads) (control, n = 8; conditional *Nedd4-2⁻/⁻*, n = 8; conditional *Nedd4-2⁻/⁻* +pirf, n = 10). **b** Volumetric analysis of fibrotic regions in the three experimental groups (control, n = 8; conditional *Nedd4-2⁻/⁻*, n = 8; conditional *Nedd4-2⁻/⁻* +pirf, n = 10). **c**, d Effect of pirfenidone treatment on static compliance (control, n = 30; conditional *Nedd4-2⁻/⁻*, n = 28; conditional *Nedd4-2⁻/⁻* +pirf, n = 27) (**c**) and on concentrations of active TGFβ in lung homogenates (control, n = 27; conditional *Nedd4-2⁻/⁻*, n = 24; conditional *Nedd4-2⁻/⁻* +pirf, n = 24) (**d**). **e**–**g** Number of neutrophils (control, n = 29; conditional *Nedd4-2⁻/⁻*, n = 25; conditional *Nedd4-2⁻/⁻* +pirf, n = 25) (**e**) and concentration of IL-13 in bronchoalveolar lavage fluid (control, n = 21; conditional *Nedd4-2⁻/⁻*, n = 24; conditional *Nedd4-2⁻/⁻* +pirf, n = 22) (**f**), and concentration of IL-1β in lung homogenates (control, n = 15; conditional *Nedd4-2⁻/⁻*, n = 15; conditional *Nedd4-2⁻/⁻* +pirf, n = 16) (**g**) of conditional *Nedd4-2⁻/⁻* and control mice treated with pirfenidone or vehicle alone. *$P < 0.05$, **$P < 0.01$, ***$P < 0.001$. **h**–**k** Effect of pirfenidone on TGFβ signaling in alveolar type 2 cells isolated from conditional *Nedd4-2⁻/⁻* and control mice induced with doxycycline for 2 weeks that were pretreated with 1 ng/ml TGFβ and analyzed for phosphorylated Smad2 (pSmad2) and transcript levels of TGFβ-regulated genes. **h** Densitometric analysis of pSmad2 at baseline and after 1–3 h of TGFβ stimulation. **i**–**k** mRNA levels of *Serpine1* (**i**), *Smad7* (**j**), and *Skil* (**k**) at indicated times after TGFβ stimulation in the presence or absence of pirfenidone. (n = 3 independent experiments). *$P < 0.05$, **$P < 0.01$, ***$P < 0.001$ compared to control mice. #$P < 0.05$, ##$P < 0.01$, ###$P < 0.001$ compared to pirfenidone treated conditional *Nedd4-2⁻/⁻* mice. Statistical analysis was performed with Kruskal wallis test and Mann–Whitney test corrected for multiple comparisons with Bonferroni method as *post hoc* test in **b**, **d**–**g**, one-way ANOVA with Tukey's post hoc test in **c** and two-way ANOVA with Tukey's post hoc test in **h**–**k**. Data are shown as mean ± S.E.M. Source data are provided in the Source Data file.

found that conditional deletion of *Nedd4-2* augmented TGFβ signaling, which resulted in hyperresponsiveness of AT2 cells to TGFβ stimulation, and increased expression of active TGFβ and pro-fibrotic marker genes in the lung (Fig. 6). This effect is consistent with lack of Nedd4-2 mediated degradation of phosphorylated Smad2/3[24] acting as important signal transducers that mediate a pro-fibrotic response. A central role of the TGFβ/Smad signaling pathway in pulmonary fibrosis was demonstrated in mice deficient for Smad3 that were protected from bleomycin induced fibrosis[42]. These data support augmentation of TGFβ/Smad signaling as an important intracellular defect contributing

to the pathogenesis of IPF-like lung disease in conditional *Nedd4-2⁻/⁻* mice. Third, based on previous reports that demonstrated a role of SP-C in familial forms of IPF[43] and recent *Sftpc*-based mouse models further supporting this notion[40,44], we determined the role of Nedd4-2 deficiency on proSP-C trafficking. Our data support a role of Nedd4-2 in cellular SP-C trafficking in vivo, but genetic deletion of *Sftpc* did neither ameliorate (e.g., due to the deletion of a potentially toxic misprocessed proSP-C), nor aggravate (e.g., due to complete loss of SP-C) pulmonary fibrosis in conditional *Nedd4-2⁻/⁻* mice (Fig. 5). These data identify misprocessing of proSP-C as an additional intracellular defect,

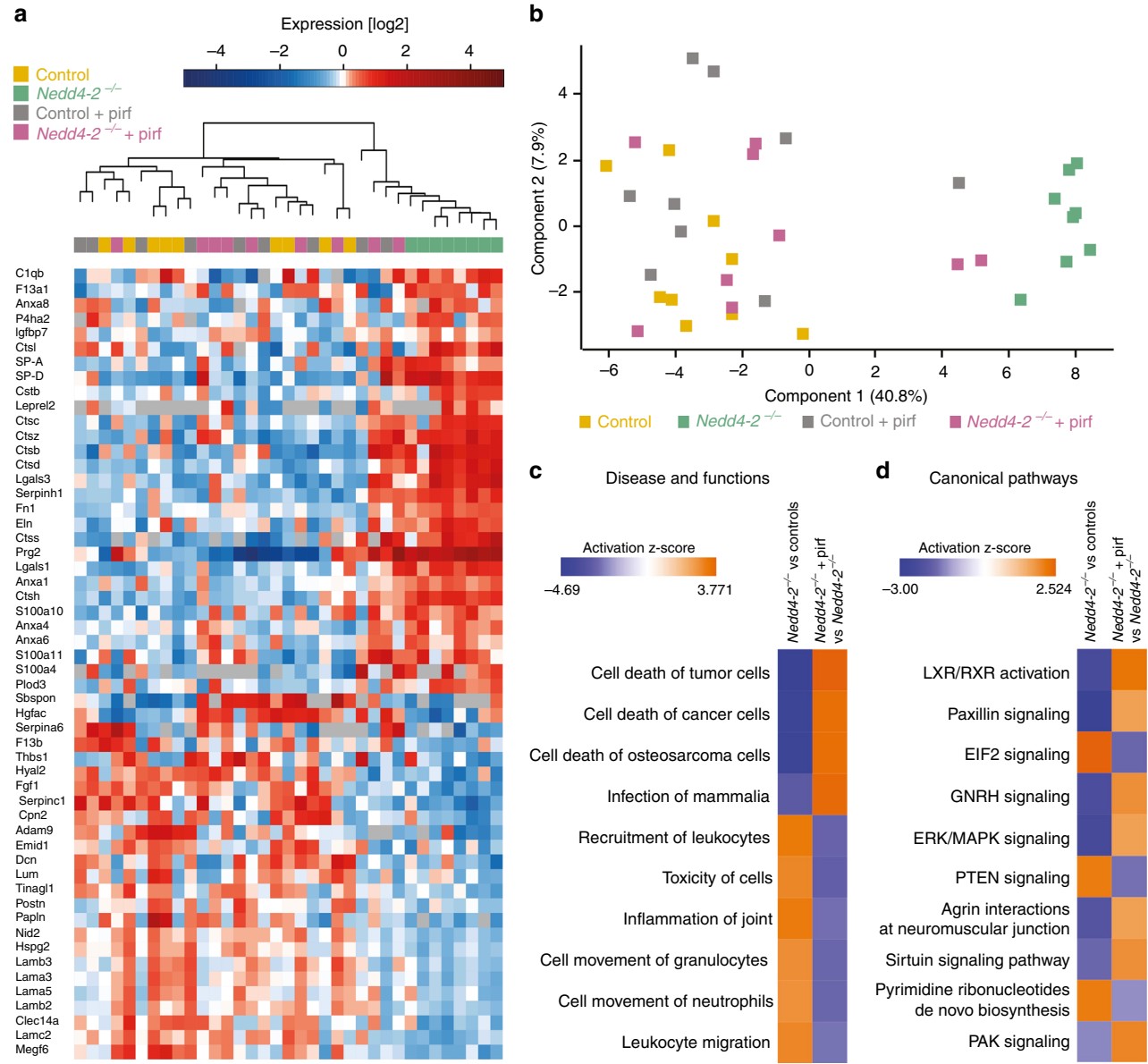

**Fig. 9 Pirfenidone treatment reverts proteomic changes associated with pulmonary fibrosis in conditional *Nedd4-2*⁻/⁻ mice. a** Heatmap of differentially regulated matrisome annotated proteins in lungs of 3-months doxycycline-induced conditional *Nedd4-2*⁻/⁻ and control mice treated with pirfenidone (pirf) or vehicle alone (*n* = 9/group and *Nedd4-2*⁻/⁻ vehicle, *n* = 8). **b** Principal component analysis of all significantly regulated proteins. **c, d** Ingenuity Pathway Analysis of significantly regulated proteins in the four experimental groups. Top enriched diseases and functions (**c**) and canonical pathways (**d**) based on the predicted activation *z*-score are shown.

but argue against a major role of this abnormality in the pathogenesis of IPF-like lung disease in conditional *Nedd4-2*⁻/⁻ mice.

Regarding the relative role and relationship of these extra- and intracellular pathogenic processes caused by Nedd4-2 deficiency, our data show that increased ENaC-mediated airway sodium hyperabsorption in freshly excised airway tissues, airway surface liquid depletion and reduced mucociliary clearance were already present after 2 weeks of doxycycline induction (Fig. 4), i.e., several months before elevated levels of active TGFβ and signs of pulmonary fibrosis were detectable in conditional *Nedd4-2*⁻/⁻ mice (Figs. 2 and 6). This sequence suggests that impaired mucociliary clearance is an early abnormality important in triggering the pathogenic cascade, likely by leading to repeated micro-injury and inflammation of the airway epithelium caused by inhaled irritants and pathogens that are retained in the lung, whereas dysregulated, hyperresponsive TGFβ signaling is an important

subsequent event that drives and perpetuates pulmonary fibrosis once epithelial micro-injury has been established in conditional *Nedd4-2*⁻/⁻ mice.

In addition to the morphological and clinical characterization of pulmonary fibrosis and studies of pathogenic processes induced by Nedd4-2 deficiency, we performed proteome analyses to provide an unbiased molecular characterization of IPF-like lung disease and compared alterations found in lungs of conditional *Nedd4-2*⁻/⁻ mice to patients with IPF. In these studies we found that a substantial number (~30%) of proteins that were differentially expressed in lung tissue of IPF patients were also differentially regulated in conditional *Nedd4-2*⁻/⁻ mice (Fig. 7). Among these overlapping dysregulated proteins, clustering analysis demonstrated a prominent role of matrisome-annotated proteins[35] indicating substantial overlap in alterations in structural ECM components that were previously identified in IPF

patients[45]. Among the proteins upregulated in lung tissue of IPF patients and conditional Nedd4-2[-/-] mice, we found COL14A1, which was reported to be expressed by a certain fibroblast subtype that undergoes expansion in bleomycin-induced fibrosis[46] and acts as a major ECM binding site for the proteoglycan decorin, which is known to regulate TGFβ activity[47,48]. Further, we found TNC as a commonly upregulated protein, which is known to be involved in cell adhesion and fibroblast migration, is transiently expressed upon tissue injury and induced by TGFβ in normal and IPF fibroblasts[49–52]. In health, TNC is activated after local tissue injury and downregulated when tissue repair or scarring are completed[50], whereas TNC production by fibroblasts was shown to be constitutively increased and correlated with excessive matrix deposition in IPF[51]. The role of TNC in promoting pulmonary fibrosis is also supported by attenuation of bleomycin-induced fibrosis in Tnc-deficient mice[52]. In addition, we found SER-PINH1, a chaperone involved in collagen maturation[53], among the commonly upregulated matrisome proteins. SERPINH1, has been shown to be increased in AT2 cells and myofibroblasts of IPF patients, where it induces overexpression of type I procollagen contributing to the general tissue disorganization[54]. Among the proteins downregulated in IPF patients and conditional Nedd4-2[-/-] mice were COL4A1, COL4A2, and COL4A3, which is expected to impede basement membrane integrity that has been suggested as possible disease initiating event in IPF[55,56] (Fig. 7). Collectively, this pattern of dysregulation of ECM proteins is consistent with increased activation of the TGFβ signaling pathway and the similarities between conditional Nedd4-2[-/-] mice and IPF patients support the relevance of this model for further studies of the complex in vivo pathogenesis of the fibroproliferative response in pulmonary fibrosis.

For an initial characterization of conditional Nedd4-2[-/-] mice as a model for preclinical testing of anti-fibrotic strategies, we determined the response to pirfenidone as approved drug for the treatment of IPF[36,37]. These studies showed that pirfenidone treatment reduced fibrotic remodeling, active TGFβ and inflammation in the lung, and improved pulmonary function in conditional Nedd4-2[-/-] mice (Fig. 8). So far, the mechanism of action of pirfenidone remains poorly understood. Previous studies in human lung fibroblasts found that pirfenidone reduced fibroblast proliferation, attenuated TGFβ-induced αSMA and procollagen I expression and inhibited TGFβ-induced Smad2/3 phosphorylation, suggesting that pirfenidone reduces fibroblast proliferation and differentiation into myofibroblasts by targeting TGFβ signaling pathways[57]. In our in vitro studies in mouse primary AT2 cells, pirfenidone abrogated increased TGFβ-induced Smad2 phosphorylation and expression of downstream target genes of TGFβ signaling such as Serpine1, Smad7, and Skil (Fig. 8). Further, at the whole organ level, our proteomics analyses show that in vivo treatment with pirfenidone reverted major proteomic changes associated with pulmonary fibrosis including dysregulation of TGFβ-induced ECM proteins in conditional Nedd4-2[-/-] mice (Fig. 8, Fig. 9; Supplementary Fig. 5). Taken together, these studies (i) demonstrate that pirfenidone slowed progression of pulmonary fibrosis in conditional Nedd4-2[-/-] mice at least in part by suppression of elevated TGFβ signaling; (ii) indicate that therapeutic benefits of pirfenidone are at least in part related to its effects on AT2 cells; and (iii) support the usefulness of this model for preclinical testing of novel anti-fibrotic compounds.

In summary, conditional deletion of Nedd4-2 in lung epithelial cells of adult mice has produced an animal model with chronic progressive pulmonary fibrosis that shares key features and molecular signatures with IPF. Further, our study shows that NEDD4-2 protein levels are reduced in patients with IPF and identifies extra- and intracellular disease mechanisms that link

Nedd4-2 deficiency to impaired mucociliary clearance and dysregulation of TGFβ signaling, two abnormalities that have been implicated in the pathogenesis of IPF[6,58]. One of the biggest challenges in IPF is that the diagnosis is often established at advanced stages of the disease. The development of pulmonary fibrosis in conditional Nedd4-2[-/-] mice is characterized by a long subclinical period followed by rapid progression of severe IPF-like lung damages at higher ages, highly similar to the disease dynamics in IPF patients. This model may therefore aid in the identification of predictive biomarkers and novel therapeutic targets, and in the preclinical evaluation of novel anti-fibrotic drugs that are urgently needed to improve the prognosis of patients with IPF.

## Methods

**Mice.** All animal studies were approved by the animal welfare authority responsible for the University of Heidelberg (Regierungspräsidium Karlsruhe, Karlsruhe, Germany). Mice with conditional deletion of Nedd4-2 in lung epithelial cells were generated by intercrossing Nedd4-2[fl/fl] mice[18], CCSPrtTA2[S]-M2 line 38[28] and LC1 mice[29]. All three lines were on a C57BL6/N background. Sftpc[-/-] mice[59] were obtained on a 129S6 background. In triple mutant Nedd4-2[fl/fl]/CCSPrtTA2[S]-M2/LC1 and quadruple mutant Nedd4-2[fl/fl]/CCSPrtTA2[S]-M2/LC1/Sftpc[-/-] mice, Cre expression is activated by the doxycycline-inducible reverse tetracycline transactivator (rtTA2[S]-M2), which is expressed under control of the CCSP promoter. To induce the conditional deletion of Nedd4-2, 4- to 6-week-old mice were exposed to 1 mg/ml doxycycline hydrochloride (Sigma) dissolved in a 5% sucrose solution supplied as drinking water in light-protected bottles. Doxycycline solutions were prepared freshly and changed at least every 3 days. For studies of the lung phenotype, 4- to 6-week-old mice were treated 3–4 months with doxycycline until clinically symptomatic or for exact periods of 0.5, 2, 3, and 4 months as indicated. Corresponding ages of mice at the time of endpoint studies were 4–5.5 month of age (for the clinically symptomatic group) or 1.5–2 month (after 0.5 months of doxycycline induction), 3–3.5 month (after 2 months of doxycycline induction), 4–4.5 month (after 3 months of doxycycline induction), and 5–5.5 month of age (after 4 months of doxycycline induction). For measurements of transepithelial ion transport and airway surface liquid height in freshly excised tracheal tissues and primary tracheal epithelial cultures, 4- to 6-week-old mice were induced with doxycycline for 2 weeks before tracheae were removed for these experiments. Mice were housed in a specific pathogen-free animal facility and had free access to food and water.

**Human lung tissues.** Paraffin embedded and cryo-preserved lung tissues from surgical lung biopsies from IPF patients and healthy tissues from biopsies of patients with pulmonary hamartoma were provided by the LungBiobank Heidelberg, a member of the Biomaterial Bank Heidelberg (BMBH) and the biobank platform of the German Center for Lung research (DZL) in accordance with the regulations of the BMBH and with the approval of the ethics committee of the University of Heidelberg (approval No. 270/2001V2-V3). Informed written consent was obtained from all patients and controls. The diagnosis of IPF was based on clinical, radiological and pathological data that were discussed at a multidisciplinary board according to current guidelines[1]. Healthy tissues from biopsies of patients with pulmonary hamartoma served as controls and were matched to the IPF cohort for age, gender, and smoking history (Supplementary Table 1).

**Pulseoxymetric measurements.** Oxygen saturation was determined at room-air using a noninvasive pulse oxymeter for laboratory animals (MouseOx Plus, Starr Life Sciences). Arterial blood oxygen saturation, respiratory rate and heart rate were recorded with a thigh clip sensor. Percent oxygen saturation was measured after stabilization of respiratory rate and heart rate.

**Pulmonary function testing.** Mice were anesthetized with sodium pentobarbital (80 mg/kg), tracheostomized and placed on the FlexiVent system (SCIREQ) for forced oscillatory measurements. Mice were then paralyzed with pancuronium bromide (0.5 mg/kg) to prevent spontaneous breathing and ventilated with a tidal volume of 10 ml/kg at a frequency of 150 breaths/min and a positive end expiratory pressure of 3 cm $H_2O$ to prevent alveolar collapse. Static compliance was determined from pressure–volume curves as described previously[60,61]. All perturbations were performed until at least three acceptable measurements were achieved.

**BAL, cell counts, and cytokine measurement.** BAL was obtained and inflammatory cell counts and macrophage size were determined as previously described[61]. Concentrations of KC (CXCL-1), IL-13, IL-1β, and active TGFβ were measured in cell-free BAL supernatant or total lung homogenates by ELISA (R&D Systems) according to manufacturer's instructions.

**Assessment of collagen content**. Collagen content of the lung was measured using the Sircol soluble collagen assay (Biocolor). Left lungs were removed and homogenized in 0.5 M acetic acid with 0.1 mg/ml pepsin and incubated over night at 4 °C. Collagen content was assessed according to manufacturer's instructions.

**Micro-computed tomography (micro-CT) imaging procedures**. For radiological characterization of the pulmonary phenotype, mice were killed by ketamine and xylazine overdose, tracheotomized, lungs were inflated with 0.8 ml of air to prevent collapse and imaged in situ with a micro-CT (Inveon, Siemens). Images were acquired with 80 kVp X-ray source voltage and an anode current of 500 µA without a filter. Exposure time was 500 ms per projection and 720 rotation steps were performed with increments of 0.5° and a total rotation of 360°. The binning factor was "1". Field of view was 29.60 × 30.06 mm producing images with an effective pixel size of 14.45 µm. Total scanning time was approximately 60 min per animal. Image reconstruction was performed using Cobra Reconstruction Software (on board software). The reconstruction filter was Shepp-Logan with a downsample factor of "2". Micro-CT images were evaluated for pulmonary fibrosis using an adapted fibrosis score[62,63]. All images were analyzed with "OsiriX" (Mac OS X, Version 5.9) with window level (WL) and window width (WW) set to 30/2500. For each animal, lungs were divided into ten levels at regular intervals along the longitudinal axis from the thoracic inlet to the dome of the diaphragm. At each level lungs were separated into right and left lung according to the perpendicular line that passes the midline of the sternum. The fibrosis score comprises subscores for (i) consolidations, (ii) reticular opacities, (iii) peripheral bronchiectasis, and (iv) parenchymal lines rated as follows. Consolidation defined as an increased attenuation with obscuration of the pulmonary vascular markings, and reticular opacities defined as a linear interstitial opacity and thickening of the peripheral connective tissue septa, were scored on a scale from 0 to 4 depending on the lung area involved (0: absent; 1: >0% to 25%; 2: >25% to 50%; 3: >50% to 75%; 4: >75%). Peripheral bronchiectasis were defined as bronchial dilatations with a maximal distance of 1 mm to the pleura and scored on a scale from 0 to 2 (0: absent; 1: >0% to 50%; 2: >50% to 100%). Parenchymal lines defined as nonvascular linear structures arising from the pleural surface and fissural thickenings were scored on a scale from 0 to 1 (0: absent; 1: non-absent). The maximum fibrosis score for the whole lung was 320. In addition, honeycombing defined by the presence of small cystic spaces with irregularly thickened walls composed of fibrotic tissue was scored separately on a scale 0–3. The score was based on the extent of fissural and pleural surface with adjacent honeycombing (0: absent; 1: >0% to 10%; 2: >10% to 20%, 3: >20%).

To quantify consolidated fibrotic regions ex vivo, inflation fixed (25 cm of fixative pressure) and paraffin embedded left lungs were scanned using a desktop small animal micro-CT (SkyScan 1176, Bruker). Images were acquired with the following parameters: 40 kVp X-ray source voltage, 600 µA current, 485 ms camera exposure time per projection, 4 projections per view and a 35 × 20 mm field of view, acquiring projections with 0.3° increments over a total angle of 180°, producing images with real pixel size of 8.65 µm. Total scanning time was 50 min. Tomograms were reconstructed using NRecon software version 1.7.0.4 (Bruker) using beam hardening correction "44%". Post alignment and ring artifact correction were optimally set for each individual scan. Reconstructed 3D datasets had an isotropic 17.3 µm voxel size and 4000 × 4000 resolution. Volumetric analysis of consolidated fibrotic regions was performed using the open source imaging analysis software Fiji[64,65]. The reconstructed image sequences were imported and images were preprocessed using a median filter to reduce background noise. Next, fibrotic areas were isolated using a threshold to select for regions with increased density and to eliminate healthy lung parenchyma. The resulting images were transformed to a binary mask. Remaining signals from small vessels were removed by a selection tool detecting for particles with a size of >400 pixels. Big vessels and artifacts were removed manually from the images if required. The remaining pixels from high density fibrotic lesions were added over the complete image sequence to determine the volume of fibrotic lung voxels.

**Histology**. Right lungs were inflated with 4% buffered formalin to 25 cm of fixative pressure. After paraffin embedding, lungs were processed for histology, sectioned at 5 µm and stained with hematoxylin and eosin (H&E) and Masson–Goldner–Trichrome. Histologic images were captured with a NanoZoomer S60 Slidescanner (Hamamatsu) using NDP.view2 software (Hamamatsu).

**Immunohistochemistry**. Lung sections of mice were evaluated for Muc5b expression using the rabbit polyclonal anti-MUC5B antibody H300 (Santa Cruz Biotechnology, SC-20119) at a dilution of 1:2000 for 2 h at 37 °C and for Muc5ac expression using the mouse monoclonal anti-MUC5AC antibody 45M1 (Novus Biologicals, NBP2-15196) at a dilution of 1:1000 for 2 h at 37 °C. To determine club cells in the lung, we performed immunohistochemistry for CCSP as a club cell marker using a rabbit polyclonal anti-CCSP antibody (Upstate, 07-623) at a dilution of 1:2000 for 16 h at 4 °C. Immunohistochemistry for tubulin was performed as a marker for ciliated airway cells using the mouse monoclonal anti-acetylated-alpha-tubulin antibody 6-11B-1 (Life Technologies, 32-2700) at a dilution of 1:1000 for 16 h at 4 °C. Extracellular matrix components were stained with rabbit anti-TNC antibody (Abcam, ab108930) at a dilution of 1:1000 for 16 h

at 4 °C, rabbit anti-COL14A1 antibody (ThermoFisher Scientific, PA5-49916) at a dilution of 1:50 for 16 h at 4 °C and rabbit anti-SERPINH1 antibody (Thermo-Fisher Scientific, PA5-27832) at a dilution of 1:200 for 16 h at 4 °C. In human lung, NEDD4-2 protein expression was evaluated in paraffin embedded tissue sections using a rabbit anti-Nedd4-2 antibody (Abcam, ab46521) at a dilution of 1:500 for 2 h at 37 °C. Unstained and hydrated paraffin sections were pretreated with 3% hydrogen peroxide in methanol followed by antigen retrieval and incubation with a nonspecific protein-blocking solution containing normal serum. For permeabilization, 0.1% Tween 20 (Carl Roth) in protein-blocking solution was used before adding the anti-MUC5AC and the anti-Nedd4-2 antibody. Tissue sections were incubated 16 h at 4 °C or 2 h at 37 °C with primary antibody. All reagents for immunohistochemistry were obtained from Vector Laboratories if not stated otherwise. As secondary antibody, we used a biotinylated goat anti-rabbit IgG (Rabbit IgG Elite) or M.O.M. mouse IgG blocking reagent followed by M.O.M. biotinylated anti mouse IgG reagent. Immunoreactivity of Muc5ac, Muc5b, and the respective cell type specific marker proteins were visualized using a peroxidase system (Vectastain Elite ABC Kit) followed by 3, 3′-diaminobenzidine (DAB) substrate kit or an alkaline phosphatase system (Vectastain ABC-AP Kit) followed by alkaline phosphatase substrate kit I. Tissue sections were treated with Avidin/ Biotin Blocking Kit before addition of a second primary antibody. As negative controls, each experiment included identically treated lung sections from controls and conditional $Nedd4$-$2^{-/-}$ mice stained with secondary antibody only. Specificity of anti-MUC5AC and anti-MUC5B antibody was tested on lung sections from IL-13 treated $Muc5ac^{-/-}$[66], $Muc5b^{-/-}$[67], and wild-type mice (kindly provided by A. Livraghi-Butrico, University of North Carolina at Chapel Hill, NC, USA) under the same conditions (data not shown). Images of immunostained lung sections were captured at a magnification of 40x with a slide scanner (Aperio AT2, Leica Microsystems).

**Immunofluorescence microscopy**. Lung sections were evaluated for proSP-C using a primary polyclonal anti-NproSP-C antibody[68,69] and Alexa Fluor 488 conjugated goat anti-rabbit IgG (Jackson Immuno Research, 111-545-062) as described previously[40]. Confocal images were acquired using a 488 nm laser line package of an Olympus Fluoview confocal system attached to an Olympus IX81 microscope (60× oil objective).

**SDS-PAGE and immunoblotting**. Sodium dodecyl sulfate polyacrylamide gel electrophoresis (SDS-PAGE) using Novex Bis–Tris gels (ThermoFisher Scientific, NP0301) and immunoblotting of PVDF membranes with primary antisera followed by species specific horseradish peroxidase conjugated secondary antisera was performed as published[40,68]. Bands detected by enhanced chemiluminescence (ECL2, ThermoFisher Scientific, 80196; or WesternSure, LI-COR, 926-95000) were acquired by exposure to film or direct scanning using an LI-COR Odyssey Fc Imaging Station and quantitated using the manufacturer's software. For immunoblotting of surfactant proteins the following antisera were used. Polyclonal anti-NproSP-C raised against the Met [10]–Glu [23] domain of rat proSP-C peptide, polyclonal anti-SP-B (PT3) raised against bovine SP-B, and polyclonal anti-SP-D (antisera 1754) raised against 2 synthetic SP-D peptides were each produced in rabbits in house and validated as published[68–71]. Polyclonal mature anti-SP-C antisera was obtained from Seven Hills Bioreagents (WRAB-76694) and validated in a prior study[40]. Monoclonal anti-ACTB was obtained from Sigma Aldrich (A1978). Unprocessed scans of the immunoblots are provided in the Source Data file.

**Morphometry**. To determine the distribution of subpopulations of airway epithelial cells including club cells, ciliated cells, and goblet cells in different regions along the tracheobronchial tree, lung sections were stained with AB-PAS or immunostained for cell type-specific markers as described above and numeric cell densities were determined using Aperio ImageScope software (Leica Microsystems) as previously described[61,72]. For each mouse, the 4 lobes of the right lung were sectioned transversally in 50 µm intervals and 4 × 10 unrelated areas within fibrotic regions were evaluated and scored for honeycomb-like and fibroblast foci-like lesions. Honeycomb-like lesions were defined as destructed lung parenchyma containing enlarged cystic airspaces surrounded by fibrotic tissue and were scored on a scale from 1 to 4 (0 = absent, 1 = 1 affected lobe, 2 = 2 affected lobes, 3 = 3 affected lobes, 4 = 4 affected lobes). The maximum score per mouse was 40. Quantification of fibroblast foci-like structures was based on observations of fibrotic consolidation with focal accumulation of fibroblasts and was scored on a scale from 1 to 4 (0 = absent, 1 = 1 affected lobe, 2 = 2 affected lobes, 3 = 3 affected lobes, 4 = 4 affected lobes). The maximum score per mouse was 40. All morphometric measurements were performed by an investigator blinded to the genotype of the mice.

**Transmission electron microscopy (TEM)**. Lungs were fixed ex situ by airway instillation using a hydrostatic pressure of 25 cm $H_2O$ with 1.5% paraformaldehyde and 1.5% glutaraldehyde in 0.15 M HEPES buffer (pH 7.35). After determination of total lung volume, a systematic uniform random sampling was performed according to standardized protocols[73]. In all, 2 mm cubes of lung tissue were then washed in 0.15 M HEPES buffer, pH 7.35 and 0.1 M cacodylate buffer, pH 7.35,

postfixed in 1% osmium tetroxide in cacodylate buffer (2 h), followed by washing steps in cacodylate buffer and water and incubation in 4% aqueous uranyl acetate at 4 °C overnight. After washing in water and dehydration in acetone, tissue cubes were embedded in Epon epoxy resin. Totally, 50 nm sections were post-stained with 4% uranyl acetate and lead citrate. Images were recorded with a Morgagni TEM (FEI) at 80 kV using a Veleta CCD camera (Olympus Soft Imaging Solutions).

**Stereologic assessment of septal wall thickness**. Lung tissues were fixed and sampled as described for TEM. Tissue slices sampled for evaluation by light microscopy were embedded in glycol methacrylate (Technovit 8100®, Heraeus Kulzer) to minimize tissue deformation[74]. A coherent test system consisting of test lines and test points was used to determine the surface density of the area of the alveolar epithelium as well as the volume fraction of the parenchymatous tissue within lung parenchyma[75]. The arithmetic mean thickness of septal walls was calculated as a volume-to-surface ratio of parenchymatous tissue[76].

**Electrogenic ion transport measurements**. Four to 6-week-old mice were induced with doxycycline for 2 weeks. Mice were deeply anesthetized via i.p. injection of a combination of ketamine and xylazine (120 and 16 mg/kg, respectively) and killed by exsanguination. Tracheal tissues were dissected using a stereomicroscope and transepithelial ion transport was measured in perfused micro-Ussing chambers as previously described[77]. Basal short circuit current ($I_{SC}$) was determined and amiloride (100 µM, luminal, Sigma) was added to assess ENaC-mediated Na+ currents as described[77].

**Airway surface liquid (ASL) height measurements**. Four to 6-week-old mice were induced with doxycycline for 2 weeks. Tracheae from ten mice per experimental group were freshly excised as described above, pooled and epithelial cells were isolated and cultured on membranes (T-Col, Costar) under air–liquid interface conditions as previously described[78]. After reaching confluence (14 days) primary tracheal epithelial cultures were washed with PBS and 20 µl of PBS containing 2 mg/ml rhodamine dextran (10 kDa; Molecular Probes) was added to the apical surface to visualize the ASL layer. Adding this volume of PBS results in an initial ASL height of 25–30 µm. Totally, 80 µl of immiscible perfluorocarbon (Fluorinert-77, Sigma-Aldrich) was added to the epithelial surface following the addition of the labeling dye as described previously[79]. Images of the Rhodamine-labeled ASL were acquired by confocal microscopy (Leica TCS SP8, Leica Microsystems) using the appropriate settings for rhodamine (excitation with 561 nm laser/emission detection at 600–650 nm). The height of the ASL was measured by averaging the heights obtained from xz scans of 16 predetermined positions on the culture. ASL height was measured at 5 min, 2 h, 4 h, 8 h, and 24 h after the addition of rhodamine dextran.

**MCT velocity measurements**. Primary tracheal epithelial cells were isolated and cultured from 6 mice per group as described above. After 14 days in culture, 5000 Nile Red carboxylate-modified fluospheres (2 µm diameter) (ThermoFisher Scientific) dissolved in 2 µl PBS were applied to the center of the apical surface of the cultures that were placed in a microscope incubator (EMBL Heidelberg) at 37 °C and 5% $CO_2$. MCT rates were determined from the bead velocity after 60 min with a Leica TCS SP8 confocal microscope (Leica Microsystems). Time lapse images were captured in 4 s intervals for 4 min at three different positions of each cell culture insert (T-Col, Costar). Image series were analyzed with the Fiji[64,65] plugin Trackmate[80] and the simple LAP tracker. Linear, directed transport tracks were identified by a ratio of track displacement to track length > 0.8, minimal displacement of 80 µm and track duration of 16 s. Mean velocity from three different positions was used to calculate a single value for each culture.

**Alveolar epithelial cell isolation**. Primary cells were harvested from conditional $Nedd4$-$2^{-/-}$ mice and littermate controls after the indicated period of doxycycline induction. Dispase (5000 caseinolytic units, Corning) was instilled through the trachea and incubated for 40 min at room temperature. After mincing the lung, the suspension was filtered sequentially through 100, 40, and 10-µm nylon meshes and centrifuged at $130 \times g$ for 8 min. The pellet was resuspended in DMEM/0.025 M HEPES. To separate macrophages, lymphocytes and endothelial cells from epithelial cells, the cell suspension was incubated with biotinylated antibodies against Cd45, Cd16/32, and Cd31 (BD Pharmingen). Obtained Cd45, Cd16/32 and Cd31 negative cells were plated on collagen-coated 24-well plates (Corning BioCoat) at 500,000 cells per well, cultured with AEC medium (SAGM Bullet Kit, Lonza) in a humidified atmosphere of 5% $CO_2$ at 37 °C and medium was changed every second day. On day 7, medium was replaced with growth factor-free medium, with or without pirfenidone (1 mg/ml; TCI) for 3 h, followed by stimulation with 1 ng/ml TGFβ (R&D Systems). Subsequently, cells were subjected to RNA or protein isolation.

**mRNA expression analysis**. Lungs from mice and surgical lung biopsies from patients with IPF and controls were either frozen with liquid nitrogen and ground with a Mikro-Dismembrator S (Sartorius) or stored at 4 °C in RNAlater (Qiagen).

Total RNA was extracted using NucleoSpin RNA Kit (Macherey-Nagel) or Trizol reagent (Invitrogen) according to the manufacturer's instructions. RNA from alveolar type 2 (AT2) cells was extracted using RNeasy Mini Plus Kit (Qiagen). cDNA was obtained by reverse transcription of 1 µg of total RNA with High-Capacity cDNA Reverse Transcription Kit (Applied Biosystems) or Superscript III RT (Invitrogen). To analyze mRNA expression of mucins and modifiers of pulmonary fibrosis in mouse lungs and $NEDD4$-$2$ expression in lung tissue from IPF patients and controls, quantitative real-time PCR was performed on an Applied Biosystems 7500 Real Time PCR System using TaqMan universal PCR master mix and the following inventoried TaqMan gene expression assays (Applied Biosystems) according to manufacturer's instructions. Relative fold changes of target gene expression were determined by normalization to expression of the reference gene $Actb$[61,81]. The TaqMan gene expression assays used for these transcript analyses are listed in Supplementary Table 2. To analyze mRNA expression of downstream target genes of the TGFβ pathway, quantitative real-time PCR was performed on a LightCycler 480 (Roche Applied Science) using the LightCycler 480 Probes Master reaction mix and hydrolysis probes (Universal Probe Library, Roche Applied Science). Crossing point values were calculated using the second-derivative-maximum method of the LightCycler 480 Basic Software (Roche Applied Science). Expression of target genes was normalized using the geometric mean of four reference genes: β-glucuronidase ($Gusb$), glucose-6-phosphate 1-dehydrogenase D ($G6pd$), glyceraldehyde-3-phosphate dehydrogenase ($Gapdh$), and hypoxanthine-guanidine phosphoribosyl transferase ($Hprt$). All primers were designed using the UniversalProbe Library Assay Design Center (Roche Applied Science). The primers used for transcript analyses by LightCycler are listed in Supplementary Table 3.

**Time-resolved analysis of Smad2/3 phosphorylation**. AT2 cells were lysed in 300 µl of whole cell lysis buffer (1% NP40, 150 mM NaCl, 50 mM Tris–HCl pH 7.4, 2.5 mM NaF, 1 mM EDTA, 0.5 mM $Na_3VO_4$, 1 mg/ml deoxycholic acid, 2 µg/ml aprotinin, and 200 µg/ml AEBSF). Whole-cell lysates were sonicated and protein yield was determined by BCA assay (Pierce). Immunoprecipitations (IP) were performed with antibodies against Smad2/3 (BD Biosciences, 610842) using protein A sepharose beads (GE Healthcare). Immunoblot analysis was performed with the primary antibodies anti-Smad2/3 (BD Biosciences, 610842, 1:2000), anti-pSmad2/3 (Cell Signaling Technology, 9520, 1:2000), anti-Actb (Sigma-Aldrich, A5441, 1:10,000), or anti-Cnx (Enzo Life Sciences, ADI-SPA-860, 1:5000) antibodies. Secondary antibodies coupled to IRDye infrared dyes (LI-COR, 926-32211 and 926-68070, 1:15,000) were used for the detection with the infrared Odyssey imager (LI-COR). Signal quantification was performed using an ImageQuant TL software (GE Healthcare). Independent replicates were scaled and averaged as described previously[82]. Unprocessed scans of the immunoblots are provided within the Source Data file.

**Mass spectrometry sample preparation and measurement**. Left lungs of conditional $Nedd4$-$2^{-/-}$ mice and littermate controls frozen in liquid nitrogen and cryo-preserved surgical lung biopsies from IPF patients and controls were ground with a Mikro-Dismembrator S (Sartorius) and lysed using 4 M urea in 250 mM $NH_4HCO_3$ supplemented with 0.1% Rapigest (Waters). Whole tissue lysates were sonicated and the soluble fraction was reduced (10 mM DTT) and alkylated (15 mM IAA). Proteins were digested with LysC (Promega) for 4 h at an enzyme to protein ratio of 1:100 followed by sample dilution to 1.5 M urea and overnight digestion with trypsin (Promega) at an enzyme to protein ratio of 1:50. Peptides were purified using stage-tips containing C18 material (Empore C18 47 mm Extraction Disc, 3 M) as described previously[83] To analyze the impact of pirfenidone on the mouse lung proteome, two lobes of the right lung were homogenized in 0.5 ml PBS with a tissue homogenizer (T10 basic ULTRA TURRAX homogenizer, IKA). The suspension was centrifuged at $10,000 \times g$ for 10 min at 4 °C. The insoluble pellets were lysed using 10 M urea in 250 mM $NH_4HCO_3$ supplemented with 0.1% Rapigest (Waters) and shaken for 30 min and combined with the corresponding soluble fractions to the final concentration of 5.6 M urea. Tissue lysates were sonicated followed by reduction (10 mM β-mercaptoethanol, Sigma), alkylation (20 mM Chloracetamide, Sigma) and quenching (10 mM β-mercaptoethanol). Proteins were digested with LysC/Trypsin mix (Promega) added at an enzyme to protein ratio of 1:25 at 4 M urea for 4 h followed by sample dilution to 1 M urea for overnight digestion. Peptides were dissolved as above. To detect NEDD4-2 peptides by parallel reaction monitoring (PRM) in cryo-preserved surgical lung biopsies from IPF patients and controls, samples were ground, lysed in 8 M urea and prepared as described for pirfenidone-treated mice. Standard peptides unique for NEDD4-2 (listed in Supplementary Table 4) labeled with $^{15}N^{13}C_6$ leucine (Peptide Specialty Laboratories (PSL)) were spiked in after peptide desalting at the amount of 2 fmol/peptide. Samples for proteome analysis were measured using an EASY-nLC 1000 (Thermo Scientific) coupled to Q Exactive Plus Hybrid Quadrupole-Orbitrap mass spectrometer (Thermo Scientific). Peptides were separated by direct injection into a 40 cm long 75 µm inner diameter column (New Objective) self-packed with ReproSil-Pur 120 C18-AQ 3 µm particles (Dr. Maisch HPLC) followed by a 5 h 0–35% acetonitril gradient. Top 20 method was applied for MS scans in an $m/z$ range of 400–1600 with a resolution of 70,000 at 200 m/z. Samples from pirfenidone treated mice and the corresponding controls were measured using the same liquid chromatography–mass spectrometry (MS) system. Peptides were separated by direct injection into a 30 cm long 75 µm inner diameter

column (New Objective) self-packed with ReproSil-Pur 120 C18-AQ 1.9 μm particles (Dr. Maisch HPLC). A 2-h gradient (100 min to 24% ACN, 20 min to 30% ACN) was used. Top 20 method was applied for MS scans in an *m/z* range of 350–1300 with an MS1 resolution of 70,000 at 200 *m/z* and an MS2 resolution of 17,500. PRM was performed on an EASY-nLC 1200 (Thermo Scientific) coupled to a Q Exactive HFX Hybrid Quadrupole-Orbitrap mass spectrometer (Thermo Scientific). Peptides were separated as described for pirfenidone treated mice with a 2 h gradient (10 min to 8% ACN, 90 min to 28% ACN 20 min to 48% ACN). A full scan at resolution of 120,000 followed by 6 DIA scans at the resolution of 30,000 targeting the 6 peptides specific for NEDD4-2 listed in Supplementary Table 4. Ion collection time was set to 400 ms.

**Mass spectrometry global data analysis**. Raw files were analyzed using Max-Quant version 1.5.8.0 with the Andromeda search engine[84,85]. Protein sequence files for human (20,000 entries) and mouse (16,000 entries) were retrieved on the 21st of December 2016 from the Swiss-Prot section of the UniProt Knowledgebase which is manually annotated and reviewed. Raw files from mouse pirfenidone treatment were analyzed using MaxQuant version 1.5.8.0. Protein sequence files for mouse were retrieved on the 16th of January 2018 from the Swiss-Prot section of the UniProt Knowledgebase. Two missed cleavage sites were allowed. Carbami-domethylation of cysteine was used as fixed modification and methionine oxidation, protein N-terminal acetylation, and hydroxyproline as variable modifications. A decoy database was created reversing original sequences. 1% false discovery rate (FDR) for both peptide and protein identification was chosen. The label-free quantification (LFQ) algorithm without normalization and minimum ratio count of 2 was applied. The resulting MaxQuant output file "protein groups" was used for protein quantification. Data pre-processing was performed in Perseus[86]. Proteins were filtered for potential contaminants, only identified by site and belonging to the decoy database. A minimum of 6 valid values was used as a threshold for protein detection. Statistical analysis was performed in the Matlab (MathWorks) framework.

The LFQ values for the protein abundances obtained by the MaxQuant search were analyzed using a data analysis pipeline comprising the following steps: i) normalization of samples, ii) averaging of technical replicates, iii) estimation of noise levels and iv) estimation of fold-changes and assessing significance. For normalization, we applied a robust version of quantile-normalization to the LFQ values, which is applicable in the case of heavy-tailed distributions (Supplementary Fig. 7). The resulting protein intensities were normalized for systematic shifts of the data distributions across samples, however, in contrast to ordinary quantile normalization, variability for high and low abundant proteins was conserved (Supplementary Fig. 6c–e). In case of technical replicates, the arithmetic mean of the replicates was calculated. To estimate biological noise levels of the data, we employed an equally weighted average of the protein-specific standard deviation SD and a prior standard deviation $SD_{prior}$ calculated from all proteins in an intensity-dependent manner. Here, locally weighted scatterplot smoothing (LOWESS)[87] with a smoothing span = 1000 was applied for calculating $SD_{prior}$ (intensity). Significance of the fold-changes between two conditions were assessed using the *t*-statistic of a multivariate statistical model[88]. Proteins with *P*-value <0.01 and <50% missing values were chosen as significantly changed. Lists of all significantly changed proteins are provided as Supplementary Data 1–3. For matrisome annotation of detected proteins, a publicly available database was used[89]. For plotting heatmaps, hierarchical clustering was applied as implemented in the amap R-package[90] using Pearson correlation for measuring similarity. Principal component analysis was performed in Perseus[86]. Gene ontology analysis was performed using the gene set regulation index (GSRI)[91], which estimates the fraction of regulated proteins for a specific gene ontology category. 50 most regulated GO terms were summarized using REVIGO[92] and merged between conditions and visualized in Cytoscape (version 3.6.1)[93]. Activation of canonical pathways and diseases and functions was analyzed through the use of IPA (QIAGEN Inc., https://www.qiagenbioinformatics.com/products/ingenuity-pathway-analysis)[94]. The MS proteomics data have been deposited to the ProteomeXchange Consortium via the PRIDE[95] partner repository with the dataset identifiers PXD011129, PXD011120, PXD011119, and PXD011116.

**MS PRM data analysis**. Raw files were analyzed using Skyline[96] version 4.1.0.18169. 6–7 fragment ions were extracted per peptide. The cumulative peak area from these transitions was used to calculate the ratio between the heavy and the light form of each peptide. Due to unreliable quantification of the SLSSPTVTLSAPLEGAK (heavy) standard peptide, this peptide was excluded from further analysis. Ratios from 2 peptides per sample were averaged. Data is shown relative to the average of the control group.

**Pirfenidone treatment**. Pirfenidone was obtained from TCI. Food with pirfenidone (0.5% w/w) and without was produced by ssniff Spezialdiaeten. Conditional *Nedd4-2*⁻/⁻ and control mice were induced with doxycycline as described above for a period of 2 months and then randomized to 2 treatment arms that received food with or without pirfenidone for a period of one month. Doxycycline induction was continued throughout the one month treatment period. At the end of the one month treatment period, pulmonary function testing was performed as described

above. Mice were killed by exsanguination and lungs were lavaged with isotonic saline (NaCl 0.9%). Differential cell count and measurement of IL-13 in BAL was performed as described above. Right lobes were removed for measurements of TGFβ and IL-1β levels by ELISA and for proteome analysis as described above. Left lungs were inflated with 4% buffered formalin to 25 cm of fixative pressure and underwent micro-CT scans after paraffin embedding as described above.

**Statistical analysis**. All data are shown as mean ± SEM unless indicated otherwise. Data analysis were performed with GraphPad Prism version 7 (GraphPad Software). Outliers were identified with Grubbs' test prior to statistical calculations. Distribution of data was assessed with Shapiro–Wilk test for normal distribution. For comparison of two groups, unpaired two-tailed *t* test or Mann–Whitney test was applied as appropriate. Comparison of more than two groups with normally distributed data sets was performed with one-way ANOVA followed by Tukey's post hoc test. Not normally distributed data were analyzed by Kruskal Wallis test and Mann–Whitney test as post hoc test with adjustment of the *P* values according to the Bonferroni method and all indicated values represent the adjusted *P* value. Survival was compared using the log rank test. A *P* value < 0.05 was accepted to indicate statistical significance.

**Reporting summary**. Further information on research design is available in the Nature Research Reporting Summary linked to this article.

## Data availability

All data supporting the findings of this study are available within the paper and its supplementary information files. The mass spectrometry proteomics data have been deposited to the ProteomeXchange Consortium via the PRIDE[95] partner repository with the dataset identifiers PXD011129, PXD011120, PXD011119, and PXD011116. For Figs. 1–6, 8 and Supplementary Figs. 1–4 raw data are provided in the Source Data file.

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

## Acknowledgements

This project was supported by grants from the German Ministry for Education and Research (FKZ 82DZL00401 and FKZ 82DZL004A1 to M.A.M., FKZ 031L0080 to C.K., FKZ 0316042A, FKZ 82DZL00404, and FKZ 82DZL004A4 to U.K., FKZ 82DZL00402 and FKZ 82DZL004A2 to F.J.F.H., M.K., and T.M.), the German Research Foundation (SPP1365/KA3423/1-1 and KA3423/3-1 to H.K. and SFB-TR84TP B08 to M.A.M.), the Canadian Institute of Health Research (CIHR MOP 130-422 to D.R.), the Canadian Foundation for Innovation (CRC, Tier I to D.R.), the VA Merit Review (1I01BX001176 to M.F.B.), the National Institutes of Health (RO1 HL119436 to M.F.B.) and the Einstein Foundation Berlin (EP-2017-393 to M.A.M.). M.F.B. is an Albert M. Rose Established Investigator of the Pulmonary Fibrosis Foundation. We thank N. Brose (Max Planck Institute of Experimental Medicine, Goettingen) for providing *Nedd4-2*[fl/fl] mice, S.W. Glasser (Cincinnati Children's Hospital Medical Center, Cincinnati) for providing *Sftpc*[−/−] mice, V. Eichwald and M. Jugold (German Cancer Research Center, Heidelberg) and W.L. Wagner (University of Heidelberg) for technical assistance and advice on micro-CT scans, A. Halavatyi (European Molecular Biology Laboratory, Heidelberg) for help in image analysis, and J. Schatterny, S. Butz, H. Scheuermann, M. Finke (University of Heidelberg), and Scott J. Russo (University of Pennsylvania, Philadelphia) for expert technical assistance.

## Author contributions

Conception and design of the study: J.D., D.H.W.L., M.S., D.D., M.O.W., M.F.B., U.K., and M.A.M. Acquisition, analysis, and interpretation of data: J.D., D.H.W.L., M.S., D.D., S.G.F., C.K., P.K.Z., A.S.A., P.K., T.E., J.H., S.M., H.K., L.K., M.O., D.R., T.M., M.K., F.J.F.H., M.O.W., M.F.B., U.K., and M.A.M. Drafting the article or revising it critically for important intellectual content: J.D., D.H.W.L., M.S., D.D., S.G.F., C.K., P.K.Z., A.S.A., P.K., T.E., J.H., S.M., H.K., L.K., M.O., D.R., T.M., M.K., F.J.F.H., M.O.W., M.F.B., U.K., and M.A.M.

## Competing interests

The authors declare no competing interests.

## Additional information

Julia Duerr [1,2,3,21✉], Dominik H. W. Leitz[1,2,3,21], Magdalena Szczygiel [2,4,5,21], Dmytro Dvornikov[2,4,5], Simon G. Fraumann[1,2], Clemens Kreutz [6,7], Piotr K. Zadora [2,4,5], Ayça Seyhan Agircan[1,2], Philip Konietzke [2,8], Theresa A. Engelmann[9,10], Jan Hegermann[9,10,11], Surafel Mulugeta[12], Hiroshi Kawabe [13,14,15], Lars Knudsen[9,10,11], Matthias Ochs [9,10,11,16], Daniela Rotin [17], Thomas Muley [2,18],

Michael Kreuter [ID][2,19], Felix J. F. Herth[2,19], Mark O. Wielpütz [ID][2,8], Michael F. Beers [ID][12], Ursula Klingmüller [ID][2,4] & Marcus A. Mall [ID][1,2,3,20][✉]

[1]Department of Translational Pulmonology, University of Heidelberg, Im Neuenheimer Feld 156, 69120 Heidelberg, Germany. [2]Translational Lung Research Center (TLRC), German Center for Lung Research (DZL), Im Neuenheimer Feld 156, 69120 Heidelberg, Germany. [3]Department of Pediatric Pulmonology, Immunology and Critical Care Medicine, Charité-Universitätsmedizin Berlin, Augustenburger Platz 1, 13353 Berlin, Germany. [4]Division of Systems Biology of Signal Transduction, German Cancer Research Center (DKFZ), Im Neuenheimer Feld 280, 69120 Heidelberg, Germany. [5]Faculty of Biosciences, University of Heidelberg, Im Neuenheimer Feld 234, 69120 Heidelberg, Germany. [6]Institute of Medical Biometry and Statistics, University of Freiburg, Stefan-Meier-Straße 26, 79104 Freiburg, Germany. [7]Centre for Integrative Biological Signaling Studies (CIBSS), University of Freiburg, Schänzlestr. 18, 79104 Freiburg, Germany. [8]Department of Diagnostic and Interventional Radiology, University of Heidelberg, Im Neuenheimer Feld 110, 69120 Heidelberg, Germany. [9]Institute of Functional and Applied Anatomy, Hannover Medical School, Carl-Neuberg-Str. 1, 30625 Hannover, Germany. [10]Biomedical Research in Endstage and Obstructive Lung Disease Hannover (BREATH), German Center for Lung Research (DZL), Carl-Neuberg-Str. 1, 30625 Hannover, Germany. [11]REBIRTH Cluster of Excellence, Carl-Neuberg-Str. 1, 30625 Hannover, Germany. [12]Pulmonary, Allergy, and Critical Care Division, Perelman School of Medicine, University of Pennsylvania, 3400 Civic Center Boulevard, 19104 Philadelphia, PA, USA. [13]Department of Molecular Neurobiology, Max Planck Institute of Experimental Medicine, Hermann-Rein-Str. 3D, 37075 Goettingen, Germany. [14]Department of Gerontology, Laboratory of Molecular Life Science, Institute of Biomedical Research and Innovation, Foundation for Biomedical Research and Innovation at Kobe, Minatojima Minamimachi, Hyogo 650-0047 Kobe, Japan. [15]Division of Pathogenetic Signaling, Department of Biochemistry and Molecular Biology, Graduate School of Medicine, Kobe University, 1-1 Rokkodaicho, Hyogo 657-8501 Kobe, Japan. [16]Institute of Functional Anatomy, Charité-Universitätsmedizin Berlin, Philippstraße 11, 10115 Berlin, Germany. [17]Hospital for Sick Children and University of Toronto, 555 University Avenue, M5G 1X8 Toronto, Canada. [18]Translational Research Unit, Thoraxklinik at University Hospital Heidelberg, Röntgenstraße 1, 69126 Heidelberg, Germany. [19]Center for Interstitial and Rare Lung Diseases, Pneumology and Critical Care Medicine, Thoraxklinik at University Hospital Heidelberg, Röntgenstraße 1, 69126 Heidelberg, Germany. [20]Berlin Institute of Health (BIH), Anna-Louisa-Karsch-Straße 2, 10178 Berlin, Germany. [21]These authors contributed equally: Julia Duerr, Dominik H.W. Leitz, Magdalena Szczygiel. [✉]email: julia.duerr@charite.de; marcus.mall@charite.de

