## [Peer Review File · Nature Communications]

Reviewers' comments:

Reviewer #1 (Remarks to the Author):

Duerr and colleagues have developed and meticulously characterized a genetically engineered mouse model of progressive lung fibrosis based on conditional knockout of the ubiquitin ligase Nedd4-2. However, novelty of the model is in question and the relationship to IPF is tenuous - beginning with the premise since the authors present no evidence for Nedd4-2 deficiency in IPF. It is essential that the authors explain why this is not just another model showing that alveolar epithelial injury leads to lung fibrosis (proof of concept published in 1981, PMID: 7315951) since it is already known that:

- Nedd4-2 deficiency causes epithelial injury and fibrosis in the kidney (PMID: 28862701)
- Nedd4-2 augments TGF- β signaling (PMID: 15496141); and activated TGF- β causes lung fibrosis.

Major points:

1. Is there evidence of Nedd4-2 deficiency in IPF? This is a critical point for motivating the study.
2. Does the model recapitulate the signature lesion in IPF- the fibroblastic focus? I found no examples of this in the manuscript as presented.

Methodological points/clarifications:

Fig 1 –

e. Specify time after induction in the fig legend. Specify how many images from how many mice the CT scans shown represent.

f, g. The authors should be specific about the n/group and the time in months to facilitate interpretation and use of the model by others. Giving ranges is not sufficient.

h. Specify the time point in the figure legend

k. What proportion of AT2 cells in each mouse showed these findings?

Reviewer #2 (Remarks to the Author):

In this study a conditional knockout of the ubiquitin ligase Nedd4-2 in the airway cells of adult mice was used to recapitulate features of IPF. While the aetiology of IPF is not resolved, the use of Nedd4-2 deletion to study epithelial dysfunction in vivo is an important step, particularly given the molecular, histological and immunological changes observed using this animal model. Further evidence for the utility of this model was obtained through measurements of ENaC expression and activity, and the activation of TGFbeta-dependent fibrosis.

The proteome analysis produced some interesting data, specifically in regards to ECM remodelling, of which a substantial number were detected in both the mouse model and IPF patients. The 'cross-validation' of the proteome between the two disease states reduces the need for orthogonal experiments to confirm the findings, although some additional immunostaining data for some of the major upregulated proteins would strengthen the conclusions. While several of these candidates were briefly discussed (COL14A, TNC increased; COL4A chains decreased), no context for these changes in terms of the current literature is provided and the reader is left wondering about their significance and how they link to the disease mechanism – are any of them known targets of TGFbeta signalling, for example?

Some aspects of the proteome analysis require more detailed explanation, as per the usual requirements of publishing large MS-based proteomics datasets (eg MCP Paris Guidelines):

"Mouse or human Swiss-Prot protein sequence files were used, respectively" – which database, version or download date and number of entries?

"The resulting MaxQuant output file "protein groups" was used for protein quantification" – were raw intensity values used or normalized (LFQ) intensity values used for quantification? If the LFQ data was used, what was the effect of a second normalization step on the distribution of LFQ values – can the authors show histograms representing pre- and post- distribution of protein intensity values.

Regarding the statistical analysis and the checklist provided, it was not entirely clear whether correction for multiple hypothesis testing was used or not, given that the authors refer to p values and not adjusted p values.

What filtering methods were used for exclusion/inclusion of proteins for statistical analysis. No mention is made of the number of proteins detected and how this data was filtered down to the list

used for statistical analysis. If proteins identified on the basis of a single peptide sequence were included, strong justification is required. What was the threshold used for inclusion of proteins with missing values (eg how many valid values were required for inclusion) and what method was used for NaN imputation?

Subject to journal requirements, I would recommend making the proteomics data publicly available via Proteomexchange and/or including supplemental tables of the MaxQuant peptides.text and ProteinGroups.txt files

Reviewer #3 (Remarks to the Author):

Nedd4-2 which is thought to regulate the epithelial Na⁺ channel (ENaC) when knocked out in mice increases expression of Muc5b in the proximal, distal, and terminal airways, and produces chronic progressive fibrosis and bronchiolization with development of honeycomb-like lesions. Treatment with pirfenidone results in diminished lung fibrosis and improved static compliance. The authors report that their data support a role for mucociliary dysfunction and aberrant epithelial pro-fibrotic response in the multifactorial pathogenesis of pulmonary fibrosis.

1. This is an important contribution because the authors present one of the few spontaneous models of fibrosis in mice that has many of the features of IPF in humans. While one could argue with some of these features (like the purported honeycomb cyst in figure 2c) the development of spontaneous model of fibrosis is an important advance.

2. More work needs to be done to understand the pathogenesis of pulmonary fibrosis in Nedd4-2 deficient mice. Neither mechanism presented in the manuscript (impaired mucociliary clearance [MCC] or regulation of TGF β signaling) to explain the development of spontaneous pulmonary fibrosis is sufficiently developed. While pirfenidone decreased the concentration of TGF β , the effect of pirfenidone on TGF β signaling and MCC was not presented. Similarly, although airway surface liquid height was presented, MCC was not directly investigated. Further definitive experiments are needed to sufficiently explore either/both MCC or TGF β signaling in this model of pulmonary fibrosis. Also, if both mechanisms are relevant, the authors will need to explain the relationship between an intra-cellular and extra-cellular pathogenic process.

3. Alternatively, figure 1K suggests that surfactant may be involved in the pathogenesis of Nedd4-2 deficient mice who develop pulmonary fibrosis. Since expression of Nedd4-2 in AT2 cells appears to be important in this model, it would be important to rule out the effect of Nedd4-2 deficiency on surfactant expression/synthesis/function.

4. Pirfenidone is likely not the best approach to demonstrating the role of Nedd4-2 in modifying the fibroproliferative response. While it's entirely unclear how pirfenidone works in IPF, treatment with

pirfenidone provides little to no help in understanding the mechanism of pulmonary fibrosis in Nedd4-2 deficient mice.

5. What is the background of the Nedd4-2^{fl/fl} mice? And is there a mixed background in the triple transgenic? If so, does this background either contribute to the phenotype or diminish the signal to noise in the model?

6. Lastly, the proteomic comparison between the Nedd4-2 deficient mice and patients with IPF was a distraction for me and didn't lead me to a mechanism. I would suggest that you further justify inclusion of these data.

Response to Reviewer 1

We wish to thank the Reviewer for the thoughtful comments and helpful suggestions. We have carefully considered all comments and performed additional experiments including assessment of NEDD4-2 levels in human IPF and a more detailed analysis of IPF signature lesions in conditional *Nedd4-2*^{-/-} mice to fully address the Reviewer's critiques and suggestions in our revised manuscript and the point-by-point response below.

Comment 1: Duerr and colleagues have developed and meticulously characterized a genetically engineered mouse model of progressive lung fibrosis based on conditional knockout of the ubiquitin ligase Nedd4-2. However, novelty of the model is in question and the relationship to IPF is tenuous - beginning with the premise since the authors present no evidence for NEDD4-2 deficiency in IPF. It is essential that the authors explain why this is not just another model showing that alveolar epithelial injury leads to lung fibrosis (proof of concept published in 1981, PMID: 7315951) since it is already known that: - Nedd4-2 deficiency causes epithelial injury and fibrosis in the kidney (PMID: 28862701) - Nedd4-2 augments TGF- β signaling (PMID: 15496141); and activated TGF- β causes lung fibrosis.

Response 1: We appreciate the Reviewer's comment on lacking evidence of NEDD4-2 deficiency in IPF in our original submission and have performed additional experiments to address this issue. Specifically, we compared NEDD4-2 expression at the mRNA and protein level in lung tissue biopsies from IPF patients and controls by i) quantitative real-time RT-PCR; ii) immunohistochemistry; and iii) quantitative mass spectrometry using parallel reaction monitoring with stable isotope labelled standard peptides. The results of these studies are summarized in the new figure 2, supplementary figure 4 and the results (page 7, para 2), and show a significant reduction (~60%) of NEDD4-2 protein and transcript levels in IPF vs control. Of note, reduced *NEDD4-2* transcript levels were observed in previous transcriptomics studies in several independent cohorts (Yang I.V. *et al.*, Thorax, 2013; Bauer Y. *et al.*, Am J Respir Cell Mol Biol, 2015), but to our knowledge, our data is the first to show that NEDD4-2 protein is reduced in patients with IPF and that genetic deletion causes progressive pulmonary fibrosis with key features of IPF in mice. We believe these data support the relevance of the conditional *Nedd4-2*^{-/-} mouse as a novel and unique model of pulmonary fibrosis and we thank the Reviewer for pointing us in this direction.

Regarding novelty, we would like to point out that in contrast to most of the published models for pulmonary fibrosis (including the proof-of-concept study mentioned by the Reviewer) that are based on exogenous administration of various toxic substances inducing acute pulmonary injury with a transient fibrotic response and mostly lack of characteristic morphological features that define IPF in patients, our data show that conditional *Nedd4-2*^{-/-} mice develop spontaneous and progressive pulmonary fibrosis with radiological and histological key features found in patients with IPF. We therefore believe that our data go far beyond of what has been published regarding animal models of pulmonary fibrosis.

Comment 2: Is there evidence of Nedd4-2 deficiency in IPF? This is a critical point for motivating the study.

Response 2: We agree this is a critical point that has been addressed with additional experiments in our revised manuscript. Please refer to Response 1 above.

Comment 3: Does the model recapitulate the signature lesion in IPF- the fibroblastic focus? I found no examples of this in the manuscript as presented.

Response 3: To address the Reviewer's question, we performed additional experiments to determine the prevalence and extent of fibroblast foci, as well as microscopic honeycombing in lungs from conditional *Nedd4-2*^{-/-} mice (the latter in response to comment 1 by Reviewer 3 below). The entire lung lobes of conditional *Nedd4-2*^{-/-} mice and controls were sectioned,

stained with hematoxylin & eosin, and fibrotic regions were scored for the presence or absence of microscopic fibroblast foci-like and honeycombing-like changes. Histological scoring of fibroblast foci was complemented by immunohistochemical staining with anti- α SMA antibody. Fibroblast foci-like changes were present in all conditional *Nedd4-2*^{-/-} mice, but were not detected in controls. On average, fibroblast foci-like changes were observed in ~45% of all lung sections evaluated. Histomorphological identification was confirmed by positive α SMA staining. These data are now included in the results (page 6) and the new figure 1 (panels j and k) of our revised manuscript. In addition, honeycombing-like changes were observed in all conditional *Nedd4-2*^{-/-} mice (non in controls) and were identified in ~27% of all lung sections evaluated (new figure 1i). We agree with the Reviewer that this more comprehensive analysis of key features found in IPF patients is important for a comprehensive description of the model and will be useful for other investigators working with this model.

Comment 4: Methodological points/clarifications:

Fig 1:

e: Specify time after induction in the fig legend. Specify how many images from how many mice the CT scans shown represent.

f, g: The authors should be specific about the n/group and the time in months to facilitate interpretation and use of the model by others. Giving ranges is not sufficient.

h: Specify the time point in the figure legend.

k: What proportion of AT2 cells in each mouse showed these findings?

Response 4: The requested information has been added to the legend of Fig. 1 as follows:

- Panels e, f, new panel g: All micro-CT imaging studies were performed after an average of 4 month of doxycycline induction at the time when conditional *Nedd4-2*^{-/-} developed clinical symptoms. For further clarification why an average induction time is reported, we also added information in the methods section in the revised online supplement stating at what age and for how long mice were induced with doxycycline, and explaining that experiments were performed when mice became symptomatic. Group sizes in these studies were n=7 conditional *Nedd4-2*^{-/-} mice and n=8 control mice and this information has been added in the legend. Panels f and the new panel g show the extent and prevalence of fibrosis and honeycomb-like abnormalities shown in the representative images in panel e.

- Panels g and h, new panels h and i: Similar to micro-CT studies, histology studies were also performed after an average of 4 month of doxycycline induction at the time when conditional *Nedd4-2*^{-/-} developed respiratory symptoms and this information along with group sizes has been added.

- Panel k, new panel o: We performed additional morphometric analyses on TEM images to quantify these findings in AT2 cells of conditional *Nedd4-2*^{-/-} mice. We found these ultrastructural abnormalities in 15.0 ± 4.5 % of all AT2 cells in lung regions that appeared macroscopically normal and in 56.1 ± 9.9 % of all AT2 cells in fibrotic regions of conditional *Nedd4-2*^{-/-} mice ($n = 3$). These data have been added to the results section of the manuscript (page 6).

Response to Reviewer 2

We thank the Reviewer for the positive evaluation and valuable suggestions to increase the potential impact of our work. We have carefully considered all comments, performed additional experiments including immunostaining of some of the major upregulated proteins in lungs of conditional *Nedd4-2*^{-/-} mice, and added further details and explanations of the proteome analyses to fully address the issues raised by the Reviewer in our revised manuscript and point-by-point response.

Comment 1: The proteome analysis produced some interesting data, specifically in regards to ECM remodelling, of which a substantial number were detected in both the mouse model and IPF patients. The 'cross-validation' of the proteome between the two disease states

reduces the need for orthogonal experiments to confirm the findings, although some additional immunostaining data for some of the major upregulated proteins would strengthen the conclusions. While several of these candidates were briefly discussed (COL14A, TNC increased; COL4A chains decreased), no context for these changes in terms of the current literature is provided and the reader is left wondering about their significance and how they link to the disease mechanism – are any of them known targets of TGF β signalling, for example?

Response 1: As suggested by the Reviewer, we performed additional immunostaining experiments for some proteins that were found to be upregulated in the lungs of patients with IPF and conditional *Nedd4-2*^{-/-} mice by our proteome analyses. Specifically we focused on Tnc, Col14a1 and Serpinh1, which were among the common differentially regulated matrix proteins and have previously been implicated in the pathogenesis of IPF. These immunostainings show increased expression of Tnc, Col14a1 and Serpinh1 in fibrotic regions of 4 months induced conditional *Nedd4-2*^{-/-} mice compared to controls and representative images are included in the new Fig. 7. Further, we have reworked the discussion to include what is known in the literature about the role of increased expression of these proteins and their link to disease mechanisms in IPF including regulation by TGF β signaling, as recommended (page 15, para 2). Further, we also added a brief discussion of the findings of reduced amounts of COL4A1, COL4A2 and COL4A3 that may contribute to loss of basement membrane integrity which has been suggested as a possible disease initiating event in IPF (page 16, para 1).

Comment 2: Some aspects of the proteome analysis require more detailed explanation, as per the usual requirements of publishing large MS-based proteomics datasets (eg MCP Paris Guidelines): "Mouse or human Swiss-Prot protein sequence files were used, respectively" – which database, version or download date and number of entries?

Response 2: We apologize for the omission and have added more detailed explanations on proteome analyses as requested. It is now detailed that protein sequence files for human (20,000 entries) and mouse (16,000 entries) were retrieved on the 21st of December 2016 from the Swiss-Prot section of the UniProt Knowledgebase that is manually annotated and reviewed. To analyze data from the mouse pirfenidone treatment study, protein sequence files for mouse were retrieved on the 16th of January 2018 from the Swiss-Prot section of the UniProt Knowledgebase. These changes have been implemented in the revised methods (page 15 in the online supplement)

Comment 3: "The resulting MaxQuant output file "protein groups" was used for protein quantification" – were raw intensity values used or normalized (LFQ) intensity values used for quantification? If the LFQ data was used, what was the effect of a second normalization step on the distribution of LFQ values – can the authors show histograms representing pre- and post- distribution of protein intensity values.

Response 3: The label-free quantification (LFQ) algorithm without normalization was applied. Afterwards, we applied the robust quantile normalization using our own scripts. This step as well as the required plots of data distribution are illustrated in the new supplementary figure 7. The additional information has been added to the revised methods (page 15 in the online supplement).

Comment 4: Regarding the statistical analysis and the checklist provided, it was not entirely clear whether correction for multiple hypothesis testing was used or not, given that the authors refer to p values and not adjusted p values.

Response 4: Correction for multiple hypothesis testing was not used and the p values provided are not adjusted p values. We used p values from a t-statistic of a multivariate statistical model with a $p < 0.01$ as significance threshold and a minimum of 50% valid

values. We did not focus on particular protein candidates but aimed at a global analysis of affected pathways and processes. Therefore we chose to work with a large, p value-determined, dataset. However, we also calculated the false discovery rate using Benjamini-Hochberg procedure and provide these in the supplementary data 1–3 next to p-values. The additional information has been added to the revised methods (page 15 in the online supplement).

Comment 5: What filtering methods were used for exclusion/inclusion of proteins for statistical analysis. No mention is made of the number of proteins detected and how this data was filtered down to the list used for statistical analysis. If proteins identified on the basis of a single peptide sequence were included, strong justification is required. What was the threshold used for inclusion of proteins with missing values (eg how many valid values were required for inclusion) and what method was used for NaN imputation?

Response 5: Data pre-processing (filtering) was performed in Perseus (Tyanova S et al., Nat Methods 2016). Proteins were filtered applying the following processing: i) for potential contaminants ii) only identified by site modification with carbamidomethylation of cysteine as fixed modification and methionine oxidation, protein N-terminal acetylation and hydroxyproline as variable modifications and iii) belonging to the decoy database that was created reversing original sequences. Proteins were not identified on the basis of a single peptide. Samples with less than 2 common peptides with other samples were excluded. A minimum of 6 valid values per protein was used as a threshold for protein detection. By filtering the initial list of 6206 hits (mouse lung), 3851 hits (human lung) and 6184 hits (mouse lung treated/untreated with pirfenidone), the number of quantified proteins was 4539 proteins from mouse lung and 2834 proteins from human lung tissues and 4565 proteins from lung tissues of pirfenidone-treated and untreated conditional *Nedd4-2*^{-/-} mice and littermate controls. To be considered statistically significant, less than 50% of missing values per protein were allowed. Missing value imputation was not performed. This additional information has been added to the revised methods (page 15 in the online supplement).

Comment 6: Subject to journal requirements, I would recommend making the proteomics data publicly available via Proteomexchange and/or including supplemental tables of the MaxQuant peptides.text and ProteinGroups.txt files.

Response 6: All proteome data from human and mouse has been uploaded on <http://www.proteomexchange.org/>. Data are available to readers via ProteomeXchange with identifiers PXD011129, PXD011120, PXD011119. For Reviewers, we provide access via credentials below concerning the 3 datasets uploaded:

Username: reviewer60950@ebi.ac.uk

Password: qMORDZgK

Username: reviewer85267@ebi.ac.uk

Password: 53jhFXDw

Username: reviewer38424@ebi.ac.uk

Password: roktHGOD

Tables of all differentially expressed proteins have been included as supplementary data 1–3.

Response to Reviewer 3

We thank the Reviewer for the positive evaluation and for the helpful comments and suggestion. We have considered all comments carefully and performed additional experiments to provide a more detailed histomorphological characterization and additional

insights into disease mechanisms to address the issues raised by the Reviewer in our revised manuscript and the point-by-point response below.

Comment 1: This is an important contribution because the authors present one of the few spontaneous models of fibrosis in mice that has many of the features of IPF in humans. While one could argue with some of these features (like the purported honeycomb cyst in figure 2c) the development of spontaneous model of fibrosis is an important advance.

Response 1: We thank the Reviewer for the comment on importance and have performed additional analyses and experiments to provide a more complete and quantitative assessment of IPF signature lesions in the lungs of conditional *Nedd4-2^{-/-}* mice. At the micro-CT level, in addition to the fibrosis score included in our original submission, all imaging data were reanalyzed and scored for radiological honeycombing-like cysts. These analysis demonstrated radiological honeycombing-like cysts in all conditional *Nedd4-2^{-/-}* mice, but none of the control mice studied (see new Fig. 1g). For a more detailed histological assessment of honeycomb-like changes as well as fibroblast foci (see comment 3 of Reviewer 1 above), we sectioned all lobes of lungs from conditional *Nedd4-2^{-/-}* and control mice, stained them with hematoxylin & eosin and scored fibrotic regions for the presence or absence of microscopic honeycombing-like and fibroblast foci-like and changes. Histological scoring was complemented by immunohistochemical staining with anti- α SMA antibody. Honeycombing-like and α SMA-positive fibroblast foci-like changes were present in all lungs from conditional *Nedd4-2^{-/-}* mice, but were not detected in controls. These data are now included in the results (page 6) and the new figure 1 (panels i-k) of our revised manuscript and provide additional support that conditional *Nedd4-2^{-/-}* exhibit these key features of IPF in humans.

Comment 2: More work needs to be done to understand the pathogenesis of pulmonary fibrosis in *Nedd4-2* deficient mice. Neither mechanism presented in the manuscript (impaired mucociliary clearance [MCC] or regulation of TGF β signaling) to explain the development of spontaneous pulmonary fibrosis is sufficiently developed. While pirfenidone decreased the concentration of TGF β , the effect of pirfenidone on TGF β signaling and MCC was not presented. Similarly, although airway surface liquid height was presented, MCC was not directly investigated. Further definitive experiments are needed to sufficiently explore either/both MCC or TGF β signaling in this model of pulmonary fibrosis. Also, if both mechanisms are relevant, the authors will need to explain the relationship between an intra-cellular and extra-cellular pathogenic process.

Response 2: We agree with the Reviewer that the question of how conditional deletion of *Nedd4-2* causes pulmonary fibrosis in this model is an interesting and important one and we thank the Reviewer for the suggestions on how to address this with additional experiments. Following the Reviewer's recommendations, we have carried out several additional experiments to further define the roles of impaired MCC and dysregulation of TGF β signaling in the pathogenesis of pulmonary fibrosis in conditional *Nedd4-2^{-/-}* mice along two main avenues.

First, to determine the effects of conditional *Nedd4-2* deficiency on MCC more directly, we cultured primary airway epithelial cells from conditional *Nedd4-2^{-/-}* and control mice and compared mucociliary transport (MCT) rates of fluorescently labeled beads on conditional *Nedd4-2^{-/-}* vs. control airway cultures. These studies showed that MCC was significantly reduced in cultures of conditional *Nedd4-2^{-/-}* vs control mice (new Fig 4d,e). When viewed in combination with our ion transport and ASL height measurements, these data are consistent with the concept that conditional *Nedd4-2* deficiency (via increased ENaC activity leading to increased transepithelial sodium/fluid absorption, ASL depletion and hyperconcentration of mucus on airway surfaces) causes impaired MCC that may in turn produce chronic epithelial injury triggering airway remodeling and fibrosis in conditional *Nedd4-2^{-/-}* mice. Interestingly this concept is supported by a recent independent study showing that mucus

hyperconcentration via overexpression of Muc5b in the airways was associated with impaired MCC and more severe and persistent bleomycin-induced pulmonary fibrosis in mice (Hancock LA et al., Nat Commun 2019).

Second, to determine the role of pirfenidone on TGF β signaling, we performed additional experiments in primary AT2 cells that were isolated from conditional *Nedd4-2*^{-/-} mice and controls and pretreated with or without pirfenidone prior to TGF β stimulation. At different time points of TGF β stimulation, AT2 whole cell lysates were analyzed for changes in phosphorylated Smad2 (pSmad2) and changes in transcript levels of downstream target genes of TGF β signaling such as *Serpine1*, *Smad7* and *Skil*. Similar to the data in figure 6 that were included in our original submission, AT2 cells of conditional *Nedd4-2*^{-/-} mice responded to TGF β with elevated levels of pSmad2 and downstream target gene expression vs. controls. This hypersensitivity to TGF β stimulation was largely abrogated after AT2 cells from conditional *Nedd4-2*^{-/-} mice were pretreated with pirfenidone (new Fig. 8h-k). In addition, we performed additional proteome analyses to determine the effect of pirfenidone treatment on the global lung proteome of conditional *Nedd4-2*^{-/-} mice (new Fig. 9 and new supplementary Fig. 5 and 6). These studies showed a reduction of multiple TGF β target genes (e.g., *Tnc*, *Serpinh1*, *Fn1*) in lungs of pirfenidone-treated vs. untreated conditional *Nedd4-2*^{-/-} mice. These results are in line with previous studies that showed inhibition of Smad2/3 phosphorylation by pirfenidone in primary human lung fibroblasts (Conte E. et al., Eur J Pharm Sci, 2014). Further, these results support a role of dysregulation of TGF β signaling in the development of pulmonary fibrosis in conditional *Nedd4-2*^{-/-} mice.

Taken together, the data of these additional experiments are consistent with the notion that both mechanisms, i.e. i) increased ENaC activity/reduced ASL height/impaired MCC; and ii) hypersensitive TGF β signaling are implicated in the pathogenesis of pulmonary fibrosis in conditional *Nedd4-2*^{-/-} mice. Regarding the relationship between an intra-cellular and extra-cellular pathogenic processes, we want to point out that airway Na⁺ hyperabsorption/ASL depletion/reduced MCC were already present after 2 weeks of Dox induction, i.e. weeks prior to the onset of pulmonary fibrosis and at a time when TGF β levels were not yet elevated (new Fig. 6a). These data support that impaired MCC is an important early abnormality that is more important in triggering the pathogenic cascade, likely by leading to repeated micro-injury of the airway epithelium and inflammation, whereas dysregulated TGF β signaling is an important secondary event that drives the development of pulmonary fibrosis once the initial airway epithelial lesions have been established in conditional *Nedd4-2*^{-/-} mice.

The additional data have been included in the new figures 4, 8 and 9 and we have reworked the results (page 8, para 2; page 11-12) and discussion (page 13-17) to include these novel data and concepts into our revised manuscript.

Comment 3: Alternatively, figure 1K suggests that surfactant may be involved in the pathogenesis of *Nedd4-2* deficient mice who develop pulmonary fibrosis. Since expression of *Nedd4-2* in AT2 cells appears to be important in this model, it would be important to rule out the effect of *Nedd4-2* deficiency on surfactant expression/synthesis/function.

Response 3: We agree with the Reviewer this is an interesting question also regarding the role of SP-C in familial forms of IPF and recent SP-C-based mouse models. To address this issue and investigate the effect of conditional *Nedd4-2* deficiency on surfactant expression/synthesis/function we performed a series of additional experiments. First, we determined protein levels of surfactant proteins in lung homogenates from conditional *Nedd4-2*^{-/-} vs control mice by Western blotting (new Fig. 5b). In these experiments, we found that the abundance of proSP-C isoforms was shifted towards the unprocessed isoform and a 16 kDa intermediate. Since SP-C biosynthesis is coupled to intracellular trafficking, we next performed immunofluorescence staining to determine the subcellular localization of proSP-C (new Fig. 5a). Cellular distribution of proSP-C was markedly changed in AT2 cells and this was associated with decreased amounts of mature SP-C in bronchoalveolar lavage fluid from

conditional *Nedd4-2^{-/-}* mice compared to controls (new Fig. 5c). This data supports the role of Nedd4-2 in cellular SP-C trafficking *in vivo*. To determine the role of this abnormality in the pathogenesis of pulmonary fibrosis, we crossed our conditional *Nedd4-2^{-/-}* mice with *Sftpc^{-/-}* mice and performed histology studies and pulmonary function testing in single (*Nedd4-2^{-/-}*) vs double (*Nedd4-2^{-/-}/Sftpc^{-/-}*) knock-out progeny (new Fig. 5e–g). These studies showed that genetic deletion of *Sftpc* in conditional *Nedd4-2^{-/-}* mice led neither to aggravation of (e.g. due to complete loss of SP-C), nor protection from (e.g. due to deletion of potentially toxic misprocessed SP-C) the development of pulmonary fibrosis. Taken together, these data show that loss of Nedd4-2 leads to proSP-C misprocessing as an additional intracellular abnormality, but that this is not a major disease mechanism underlying the pathogenesis of IPF-like lung disease in conditional *Nedd4-2^{-/-}* mice. The data of these additional experiments are included in the new figure 5, the results (page 9, para 2) and discussion (page 14, para 1) of our revised manuscript.

Comment 4: Pirfenidone is likely not the best approach to demonstrating the role of Nedd4-2 in modifying the fibroproliferative response. While it's entirely unclear how pirfenidone works in IPF, treatment with pirfenidone provides little to no help in understanding the mechanism of pulmonary fibrosis in conditional *Nedd4-2^{-/-}* mice.

Response 4: The primary purpose of the pirfenidone treatment study was not to understand the mechanism of pulmonary fibrosis in conditional *Nedd4-2^{-/-}* mice, but rather to test if pirfenidone as one of the first approved drugs for IPF patients has effects on the development of IPF-like lung disease in this model that may then serve as a reference for future preclinical testing of emerging anti-fibrotic compounds. We agree with the Reviewer that the mode of action of pirfenidone remains not well understood. However, there are data showing that pirfenidone has inhibitory effects on fibroblast migration and proliferation and also reduced secretion of TGFβ from fibroblasts *in vitro* (Lehtonen S.T. et al., *Respir Res.*, 2016; Conte E. et al., *Eur J Pharm Sci*, 2014). Further, pirfenidone was the first approved drug that was able to reduce clinically relevant outcomes in IPF patients, albeit we agree that effect sizes are modest and at the expense of substantial side effects. In our additional experiments performed in response to the Reviewer's comment 2 above, we found that pirfenidone largely abrogated the observed hypersensitivity of *Nedd4-2* deficient AT2 cells to TGFβ stimulation both at the level of signal transduction, as well as target gene expression (new Fig. 8h–k). Further, we show that pirfenidone treatment reduced active TGFβ and severity of pulmonary fibrosis in conditional *Nedd4-2^{-/-}* mice. Finally, additional analyses of the effect of pirfenidone treatment on the lung proteome demonstrated a reduction of multiple TGFβ target proteins (Tnc, Serpinh1, Fn1) as well as global changes of the proteome in conditional *Nedd4-2^{-/-}* mice (new Fig. 9 and suppl fig 5 and 6). Therefore, in addition to our response to the Reviewer's comment 2 above, we think that the pirfenidone treatment studies support that this new model with IPF-like lung disease is useful for preclinical testing of approved and novel antifibrotic agents that will hopefully accelerate the develop of more effective therapies for patients with IPF.

Comment 5: What is the background of the *Nedd4-2fl/fl* mice? And is there a mixed background in the triple transgenic? If so, does this background either contribute to the phenotype or diminish the signal to noise in the model?

Response 5: All 3 transgenic lines were backcrossed onto the C57BL6/N background before they were intercrossed to generate the triple transgenic conditional *Nedd4-2^{-/-}* mice. We can therefore exclude contributions of different genetic backgrounds to the variability of the lung phenotype observed in the triple transgenic conditional *Nedd4-2^{-/-}* mice. This information on strain backgrounds has been added to the revised methods (page 2, para 1 in the online supplement)..

Comment 6: Lastly, the proteomic comparison between the conditional *Nedd4-2^{-/-}* mice and

patients with IPF was a distraction for me and didn't lead me to a mechanism. I would suggest that you further justify inclusion of these data.

Response 6: The main purpose of the proteomic analyses was to provide an unbiased molecular characterization of IPF-like lung disease at the level of the lung proteome, and a comparison to alterations found in patients with IPF, as part of a comprehensive initial description of the conditional *Nedd4-2*^{-/-} mouse. In our revision, these analyses were extended to treatment effects of pirfenidone on global alterations of the proteome and changes in TGFβ regulated target genes. We believe that this characterization including identification of commonly dysregulated proteins, pathways and biological functions, and response to therapeutic intervention with an approved IPF drug, provides useful information that will also be helpful for other investigators and further studies of the complex *in vivo* pathogenesis and preclinical testing of novel anti-fibrotic strategies in this new model with IPF-like disease. We acknowledge that due to the brevity of the previous letter format, justification of this data was limited and we have reworked the results (page 11, para 2) and discussion of our revised manuscript (page 15, para 2) to clarify the purpose and relevance of our proteomic analyses (see also response to comment 1 by Reviewer #2).

REVIEWERS' COMMENTS:

Reviewer #1 (Remarks to the Author):

The authors have addressed my concerns. As an optional suggestion, I think the manuscript would flow better if Figure 2 preceded Figure 1.

Peter Bitterman

Reviewer #2 (Remarks to the Author):

The authors have made a substantial improvement to this manuscript and I am satisfied with their approach to addressing my comments. In particular the new data that validates the results of the proteomics analysis is convincing and the justification for the statistical criteria used to cast a broad survey for proteins of interest is reasonable.

Reviewer #3 (Remarks to the Author):

The authors have sufficiently addressed all of the concerns raised in my previous review of the manuscript. My only remaining minor suggestion is to change MUC2 on line 178 to MUC5B. Variants in MUC2 have never been shown to be genetically related to IPF.

David A. Schwartz

Response to Reviewer 1

Comment 1: The authors have addressed my concerns. As an optional suggestion, I think the manuscript would flow better if Figure 2 preceded Figure 1.

Response 1: We thank the Reviewer for this suggestion and have changed the sequence of the figures and corresponding text in the results and first paragraph of the discussion accordingly.

Response to Reviewer 2

Comment 1: The authors have made a substantial improvement to this manuscript and I am satisfied with their approach to addressing my comments. In particular the new data that validates the results of the proteomics analysis is convincing and the justification for the statistical criteria used to cast a broad survey for proteins of interest is reasonable.

Response 1: We thank the Reviewer for this positive feedback and for pointing us in this direction.

Response to Reviewer 3

Comment 1: The authors have sufficiently addressed all of the concerns raised in my previous review of the manuscript. My only remaining minor suggestion is to change MUC2 on line 178 to MUC5B. Variants in MUC2 have never been shown to be genetically related to IPF.

Response 1: We thank the Reviewer for spotting this and have changed the text as suggested.